# Modeling intrinsic factors of inclusive engagement in citizen science: Insights from the participants' survey analysis of CSI-COP

Shlomit Hadad[1]*, Maayan Zhitomirsky-Geffet[2], Huma Shah[3], Dorottya Rigler[4], Ulrico Celentano[5], Henna Tiensuu[5], Juha Röning[5], Jordi Vallverdú[6], Eva Jove Csabella[6], Olga Stepankova[7], John Gialelis[8], Konstantina Lantavou[8], Tiberius Ignat[9], Giacomo Masone[3], Jaimz Winter[3], Marica Dumitrasco[10]

1 Department of Digital Learning Technologies, The Israel Academic College in Ramat-Gan, Raanana, Israel, 2 Department of Information Science, Bar-Ilan University, Ramat Gan, Israel, 3 Department of Computational Science and Mathematical Modelling, Coventry University, Coventry, United Kingdom, 4 Independent Researcher, Hungary, 5 Biomimetics and Intelligent Systems Group, University of Oulu, Oulu, Finland, 6 Philosophy Department, Universitat Autònoma de Barcelona (UAB), Barcelona, Spain, 7 Department of Biomedical Engineering and Assistive Technology, Czech Institute of Informatics, Robotics and Cybernetics, The Czech Technical University in Prague, Prague, Czech Republic, 8 Electrical and Computer Engineering Department, University of Patras, Patras, Greece, 9 Immer Besser GmbH and SKS Knowledge Services, Munich, Germany, 10 Department of Science and Innovation Management Academy of Music, Theatre and Fine Arts, Chisinau, Republic of Moldova

☯ These authors contributed equally to this work.
‡ HS, DR, UC, HT, JR, JV, EJC, OS, JG, KL, TI, GM, JW and MD also contributed equally to this work.
* shsh3345@iac.ac.il

**Data Availability Statement:** All relevant data for this study are publicly available from the Zenodo repository (https://zenodo.org/records/10043687).

## Abstract

Inclusive citizen science, an emerging field, has seen extensive research. Prior studies primarily concentrated on creating theoretical models and practical strategies for diversifying citizen science (CS) projects. These studies relied on ethical frameworks or post-project empirical observations. Few examined active participants' socio-demographic and behavioral data. Notably, none, to our knowledge, explored prospective citizen scientists' traits as intrinsic factors to enhance diversity and engagement in CS. This paper presents a new inclusive CS engagement model based on quantitative analysis of surveys administered to 540 participants of the dedicated free informal education MOOC (Massive Open Online Course) 'Your Right to Privacy Online' from eight countries in the EU funded project, CSI-COP (Citizen Scientists Investigating Cookies and App GDPR compliance). The surveys were filled out just after completing the training stage and before joining the project as active CSs. Out of the 540 participants who completed the surveys analyzed in this study, only 170 (32%) individuals actively participated as CSs in the project. Therefore, the study attempted to understand what characterizes these participants compared to those who decided to refrain from joining the project after the training stage. The study employed descriptive analysis and advanced statistical tests to explore the correlations among different research variables. The findings revealed several important relationships and predictors for becoming a citizen scientist based on the surveys analysis, such as age, gender, culture, education, Internet accessibility and apps usage, as well as the satisfaction with the MOOC, the mode of training and initial intentions for becoming a CS. These findings lead to the development

**Funding:** This communication is part of a project that has received funding from the European Union's Horizon 2020 research and innovation program (under grant agreement N˚873169). Initials of the authors who received each award; H. S. The funders had no role in study design, data collection and analysis, decision to publish, or preparation of the manuscript.

**Competing interests:** The authors have declared that no competing interests exist.

of the empirical model for inclusive engagement in CS and enhance the understanding of the internal factors that influence individuals' intention and actual participation as CSs. The devised model offers valuable insights and key implications for future CS initiatives. It emphasizes the necessity of targeted recruitment strategies, focusing on underrepresented groups and overcoming accessibility barriers. Positive learning experiences, especially through MOOCs, are crucial; enhancing training programs and making educational materials accessible and culturally diverse can boost participant motivation. Acknowledging varying technological proficiency and providing necessary resources enhances active engagement. Addressing the intention-engagement gap is vital; understanding underlying factors and creating supportive environments can transform intentions into active involvement. Embracing cultural diversity through language-specific strategies ensures an inclusive environment for effective contributions.

## Introduction

Citizen science has gained popularity in the past decade, allowing the general public to participate in voluntary work that benefits citizen scientists and adds research value. Involving non-professionals in data collection increases scientific literacy and awareness of relevant issues. Citizen science (CS) projects have the potential to empower individuals from diverse backgrounds to contribute to scientific research [1, 2]. However, ensuring inclusivity in CS is crucial, as power disparities and partnership challenges can limit participation [3, 4]. Studies have revealed variations in response rates, with some projects receiving a high percentage of responses while others face challenges in engaging participants [5, 6]. Addressing these challenges requires considering the roles of participants, compensation for their work, usage of technology and the need for sustained funding [7–9]. Effective communication and project design are also key in increasing participation and inclusivity [10]. To enhance inclusivity, projects should focus on relevant research, create positive experiences, consider participant motivations, utilize volunteer-centric frameworks and digital technology [11–13].

In addition, understanding the demographic composition of CS projects is essential for promoting inclusivity [14]. Demographic data reveals disparities in race, disabilities, gender, education levels, and age groups [15, 16]. Collecting demographic data in an ethical manner is essential for assessing project inclusivity and understanding engagement factors in CS projects [5]. Therefore, ethical considerations and data protection regulations should be followed to collect and analyze demographic data, allowing for a comprehensive understanding of the project inclusivity and factors influencing engagement in CS projects [2, 10].

While numerous recent studies address the topic of inclusivity in CS, only few of them collected socio-demographic and behavioral data of the citizens involved in a CS project to analyze intrinsic factors that may help increase engagement and diversity of citizens in CS. Moreover, most of the abovementioned studies analyzed the data of active citizen scientists after the completion of the project. To the best of our knowledge, this is the first research that examined participants' socio-demographic, behavioral, intention- and training-related data collected in the middle of the recruitment process to model the major intrinsic characteristics of actively engaged citizen scientists compared to passively interested participants who left the project after the training stage.

The CSI-COP (Citizen Scientists Investigating Cookies and App GDPR compliance) project, aimed at investigating the General Data Protection Regulation (GDPR) compliance and

privacy issues, engaged citizen scientists to address growing concerns about privacy in society [17]. The project consortium included leading academic institutions and private companies from Europe and Israel: the lead and coordinating partner Coventry University (UK), University of Patras (Greece), University of Oulu (Finland), Bar-Ilan University (Israel), Czech Technical University in Prague (Czech Republic), NaTE Association of Hungarian Women in Science (Hungary), Autonomous University of Barcelona (Spain), Immer Besser and Stelar (Germany). However, not all partners were involved in the recruitment and informal education of citizen scientists. Data protection and privacy legal expert, Stelar were involved in reviewing CSI-COP participant information to ensure the project complied with the GDPR. To ensure inclusivity, the project adopted the ten principles of citizen science inclusion formulated by ECSA (European Citizen Science Association, 2015).

Calls for participation were announced on citizen science platforms, social media channels, and through direct letters to relevant organizations and press releases. Citizen scientists were informally educated about data protection and privacy rights under the GDPR, receiving practical training on uncovering cookies in websites and apps through workshops using a dedicated MOOC developed by the project's coordinating partner (Coventry University), and its sub-contractor (Privacy Matters). The MOOC was translated by the partners to deliver it to their citizens in the local language. The CSI-COP project stands out due to its extensive training and learning requirements for citizen scientists. Participants were required to undergo a comprehensive MOOC course, which involved approximately 2 ½-3 hours of self-learning or attending a MOOC-based workshop given by one of the partners of similar duration. The project involved collecting information about trackers on websites and apps, which entailed examining cookie notices and privacy policies on websites, and permissions in apps. Citizen Scientists were made aware of free online privacy audit tools to utilize for tracker identification and analysis, and recording their findings in a detailed bespoke Excel table with various fields, including relevant screenshots for validation [18]. In particular, by complying with GDPR so obtaining informed consent from interested participants, CSs were involved in co-investigating the use of cookies of various types in websites and trackers in their mobile devices.

Hence, compared to typical citizen science projects, CSI-COP project demanded significantly more time, motivation, concern about online personal data collection, knowledge, and investment from the participants, in addition to their involvement in form-filling and survey completion for research purposes. These unique challenges presented obstacles in recruiting participants, further compounded by the difficulties posed by the COVID-19 pandemic, as noted in a CSI-COP forthcoming paper [19], and other studies [20]. The combined efforts of the CSI-COP team and volunteer citizen scientists resulted in the creation of a Taxonomy of Cookies and Online Trackers [19], and an open-access, searchable repository of trackers found from CSI-COP investigations. The Repository is available on the CSI-COP project website (https://csi-cop.eu/repository/). Stakeholders can explore this database to determine if any trackers are present on the web pages they visit.

In the current study, we present findings from quantitative analysis of surveys administered to 540 learners of the dedicated MOOC from eight countries in the EU funded CSI-COP project just after successfully completing the training stage and before joining the project as citizen scientists. Only 170 (32%) of the 540 participants who completed the surveys have become citizen scientists in the project.

The following research questions were examined in the study:

RQ1: Are there relationships between the *socio-demographic* variables, the *frequency of apps usage on computers/mobile*, *satisfaction from the training (MOOC)*, *mode of course completion* and the *behavioral intentions to become a CS* and *becoming a citizen scientist*?

RQ2: Is there a difference in the level of intention to join the project as citizen scientists between participants who became citizen scientists and those who did not?

RQ3: Does the language of the participants (Greek, Romanian, Czech, English, Hebrew, Hungarian, Spanish) influence the study variables?

RQ4: Is there an effect of the different modes of training (self-training via MOOC, online workshop, face-to-face workshop) on the training satisfaction level, the intention of participants to join CSI-COP as citizen scientists, and their actual involvement as citizen scientists in practice?

RQ5: Which of the variables examined in the study predicts the actual participation as a citizen scientist?

## Literature review

### Fostering inclusiveness and ethical considerations in citizen science

In recent years, the topic of inclusiveness and diversity in citizen science (CS) has gained increasing attention. The 9th principle of the ECSA's Ten Principles of Citizen Science reads "Citizen science programmes are evaluated for their scientific output, data quality, participant experience and wider societal or policy impact" [21]. It makes clear that participant experience is central to CS and there is a need for enhanced inclusivity in CS [22]. Such an experience depends on the demographics and typology of volunteers. Researchers have conducted studies to understand the current situation and propose strategies to increase the participation and engagement of diverse citizen scientists. However, to the best of our knowledge, no previous CS project engaged citizen scientists in investigating information technology and/or its ethical, societal and judicial aspects as done in this research.

It is important to recognize and address the moral-relational complexities in CS [8]. Rasmussen and Cooper [23] note the distributed nature of practice across disciplines and the need to engage members in co-creating ethical norms. Fiske et al. [10] propose a series of questions for planning, operation, funding, participation, and ethical supervision of CS projects. These queries cover participation conditions, obstacles and burdens, benefits and dissemination of results, representation, acknowledgment, trust, risk, and disparities. Integrating these guidelines at the project design stage, rather than as afterthought limitations, would avoid situations where citizen scientists feel excluded, unsafe, or violated during research [9]. CS should not exploit individuals for free labor [2], but instead strive for collaboration, co-ownership, democratic decision-making, data sharing, intellectual property, and exploitation [9].

### Harnessing digital technologies for inclusive participation in citizen science

Social inclusivity is perceived as a cornerstone of CS [24]. Inclusive CS can empower participants to contribute to science, regardless of their educational background, literacy skills, or cultural origins, dismantling barriers between professional scientists and citizen scientists, thereby expanding the scope and capability of the scientific community [25].

The advent of accessible digital technologies has facilitated active and inclusive grassroots participation, by improving communication, and internal coordination between experts and volunteers [8]. Digitally-facilitated CS research aided recruitment and increased diversity in the study population [14]. Paleco et al. [4] suggest considering training in technology use, also providing incentives and career opportunities for young researchers. Technology and more widely literacy and ability barriers must be addressed. Moustard et al. [11] observed that rural

communities and local participants may be non-literate but are most ecologically literate. Their involvement was facilitated through use of a smartphone application based on Sapelli, an open-source configurable icon-driven user interface for data collection across language and literacy barriers [12].

Similarly, a 1996 project used an icon-based user interface for handheld computers, allowing trackers to document intricate biodiversity data, also demonstrating that scientific reasoning might be an inherent human capability [26]. Enabling the use of participants' own languages overcomes barriers and leads to diverse participation [4] in more general cases where graphics are not enough or applicable. Offering online participation options can enhance diversity as well [14]. Similarly, for recruitment and engagement. At least for young participants (18–29 years old), it is could help to provide social opportunities through platforms like WhatsApp, Slack, and Yammer [27]. On the other hand, more traditional approaches, such as word-of-mouth, may help reach other individuals [25]. It is to be noted that at the project-level CSI-COP did not use Meta owned apps (Facebook, Instagram, WhatsApp) due to data protection issues reported widely in the press about Facebook, such as in the case of Cambridge Analytica's use of personal data from that social media platform.

For diversity, it is important to improve communication and raise awareness [25]. Communication and interaction may even trigger CS activities, as with discussion on a Finnish Facebook group on citizens' observation of auroras, where interaction between citizens and academics led to discovery of a novel aurora type, then featured in a joint academics-citizens scientific article [11]. CS may have broader impact on society, for example, Diblíková et al. [28] noted that more than half of studied participants cited the use of iNaturalist app as the reason for more frequent trips into nature.

## Addressing roles, inequalities, and communication in citizen science

CS implies more than just contributing to research; it requires some knowledge of the field, its methods and literature [2]. CS often involves power disparities and partnership challenges between scientists and non-academics, with feeling of inequality, a perception of scientific knowledge superiority [8], or challenges in data interpretation, long-term participation, and specific language requirements [4] that can limit inclusiveness. The above boils down to the issue of roles in the CS project. Among the solutions, it is recommended to diversify citizens' roles [27], not limiting their role to citizen scientists, but raise their involvement up to stakeholders or partners [29]. This brings two challenges. The already discussed inequalities between researchers and citizen scientists include the exclusion of citizen scientists from project benefits, including funding for jointly produced research [8] and the unpaid aspect of volunteering and the costs of participation, including transportation and equipment may represent a barrier [27]. It is then suggested that compensation for their work could be planned in the project [29].

There is a wider problem, the fact that within the broader scientific funding context there is often little opportunity to include stakeholders at project conception and during its development phases [29]. This is unfortunate, since dialogue between different stakeholders and enabling non-experts to articulate their views on community interests and local issues, may bolster social inclusiveness and equip participants with knowledge and tools necessary for tackling problems [24]. Funding limitations also have long-term effects. Liebenberg et al. [25] report that, while the project proved successful–with pride and empowered trackers, and both trackers and scientists reporting mutual learning and benefits–the lack of sustained funding causes initiatives to be ephemeral and with limited socio-economic benefits. Ensuring sustainability, CS projects in developing nations will require international support [25].

Various suggestions and recommendations for increasing participation and inclusivity have been derived. They basically condense into communication and project design requirements. Effective communication and dissemination strategies in CS projects are crucial in increasing visibility, engaging participants, and influencing policy, but there is the need of tailoring communication initiatives to the target audience [30]. Considering opinions and ideas from online group discussions improves inclusivity [27]. It has been noted that addressing citizens through Facebook groups facilitated interactions and engagement [26], but creative approaches such as storytelling and non-written communication tools are also suggested to engage potential participants [30].

## Designing citizen science projects for inclusivity, engagement, and sustained motivation

To enhance inclusivity, CS project design should properly address both the topic and the methodology. The project should target positive social change, focus on answering relevant research questions [22] and target creating tangible impacts [27], but topics should also appeal participants with different levels of preparedness [26] and recognize the necessities of those underrepresented [22]. To boost engagement, the project should ensure a positive experience and a welcoming forum, and by sharing enthusiasm, ideas, and knowledge, it should provide learning opportunities [26]. Understanding the motivation for joining as a function of participant identity is a fundamental part of intentional project design [13].

Looking at psychological theories, Palacin et al. [31] highlight the key values that underpin initial and sustained motivations as self-direction, stimulation, achievement, and security, with additional values including universalism, benevolence, conformity/tradition and power. Since participants who persisted past observation gained in their ability over time and the quality of their data was growing [13]. Nurturing sustained motivation is important for both sides of the CS project. Parrish et al. [12] show that online classification tasks produced a large number of results but a short-term involvement, whereas deductive hands-on activities are associated with lesser contributions but longer-term dedication. When dealing with a diverse population of citizens, outcomes may differ even within the same project, e.g., Korkala [32] observed that most participants may contribute just one or two observations, with few participants being very active.

## Examining the demographic composition of citizen scientists: Insights from multiple studies

Discussing inclusivity and means to improve participation and diversity obviously requires looking at demographics and other characteristics of citizens that engage in activities. Availability of those data is not wide and not even. Moczek et al. [14] conducted a survey among 140 CS projects in Germany to gather information about the involvement of academic researchers and citizen scientists. The results revealed that while a majority (61.5%) of projects collected personal data from citizen scientists, including contact details, but only about a quarter of the projects recorded age, gender, or place of residence (25.6%, 24.4% and 24.4%, respectively) and even fewer projects (14.1%) asked for affiliation to target groups (e.g. nationalities, ethnicities, income, occupational groups). The authors suggest that systematically collecting personal characteristics could potentially break trust and discourage potential participants from getting involved. This lack of information on participants' demographics and diversity is also typical for projects in most European countries, which indicates an uncertainty among projects about appropriate methods for collecting such data.

Response rate at data collection varied in different studies. Thus, Moczek et al. [14] got answers from 65 projects out of 140 (46%), with data about 63,339 citizen scientists in total.

The Conservation Volunteers (TCV 2014) UK survey received 19,256 responses out of 38,137 (51%). Korkala [32] got survey answers from 188 out of 609 enquired (31%). In Caltová et al. [5] only 9% of 94 participants responded, possibly because a broad community of users with varying involvement was targeted. Some studies reported on basic demographic traits of the citizen scientists. The citizen scientist in USA projects or in international activities with majority of participants from the USA were predominantly white, also compared with the ratio in US population [33], observation aligned with the one in Cooper et al. [22]. The UK analysis (TCV 2014) noticed a predominance in white people, with people non-white and from UK or Irish represented at 23%. Information on disabilities is rarely recorded, but in the UK survey [34], people with disabilities were only 9% of participants, against 18% of total UK population according to 2011 census.

Gender information: In an analysis by Sprinks et al. [35] on Zooniverse Planet Four: Craters astronomy platform ($n = 30$), 63% of the participants were male and 37% female. In TCV (2014) UK survey ($n = 19256$), females were 51%. In a Czech Republic study [5], most common participants were women (62%, $n = 94$). In Israel, women are active citizen scientists [16]: in an air pollution monitoring project, participants ($n = 131$) were equally divided between men and women, but in [36] 57% ($n = 123$) were female.

Citizen scientists in USA projects or in international activities with majority of participants held advanced degrees [22, 33]. Similarly, the UK [34] tells a predominance in educated people amongst adult participants. Along the same lines, all participants in Planet Four were university educated [35] and most participants in the Czech study [5] had a university degree, aligned with the findings for Israel in Golumbic et al. [15] and Soleri et al. [16]. Finer classifications are found in a few studies. Golumbic et al. (2020) report for Israel varying education levels, with 5% having no degree, 45% a Bachelor's, 45% a Master's, and 5% a PhD ($n = 123$). Korkala [32] found for Finnish Mushroom Atlas ($n = 188$) 3% with lower secondary education, 10% with vocational school degree, 8% with high school or upper secondary degree, 15% with post-secondary school education, 16% with university of applied sciences degree, and 27% with university education.

For age groups data granularity is uneven. The TCV [34] survey reported that 25,329 participants (66%) were adults and 12,808 (34%) children. Sprinks et al. [35] reported that the age of candidates was between 22 and 60 years, centered around 28 years. The Israel air pollution project had participants in the range 20–70 of Soleri et al. [16], similar to another Israel project showing age distribution ranged 18–70, with the highest proportion in the 31–50 range [36], but the opposite was noted for the Israel butterfly project, where the largest age group (33%, $n = 44$) was 65 and older [37]. Despite the use of a mobile application, the main category of Czech respondents ($n = 94$) was 31–40 years (42%) and 41–50 years old (28%), other age groups were represented by less than 10%, with the youngest respondents 12–15 years old, the oldest over 71 years [5]. For Finnish Mushroom Atlas [32] the age was in the wide range 14–75, with an average of 50. Age groups are not specified there, but the duration of mushroom hobby is registered: 0–10 years 27%, 11–20 years 17%, 21–30 years 17%, 31–40 years 16%, 41 or more 23% [32].

The above studies bring to evidence two more factors worth noting to enhance engagement and inclusivity. [33] observe that approximately 77% of participants engage in multiple projects, therefore they propose a volunteer-centric framework for engaging volunteers across multiple projects and modes to promote inclusivity. Komai et al. [37] report that only 25% of participants used mobile apps, with reasons for non-use including perceived lack of advantage or technical difficulties. Concluding this section, articles reviewed for this paper shed light on the importance of inclusiveness and diversity in CS projects. They highlight the need for appropriate methods to collect demographic data, the significance of effective communication

and dissemination strategies, the value of volunteer-centric frameworks, and the general potential of digital technology to enhance participant recruitment and diversity. By implementing these recommendations, CS projects can work towards creating more inclusive and diverse research communities [38]. Hence, in CSI-COP project we complied with responsible research and ethical principles as well as with the EU's data protection regulation (GDPR) to anonymously collect and analyze the personal, behavioral, and socio-demographic data of the MOOC learners and active citizen scientists. This allows us to learn about the inclusivity of the project and determine the influence factors of active inclusive engagement in CS projects.

## Method

### Citizens' recruitment ethics and engagement methodology in CSI-COP

The CSI-COP project, which focuses on GDPR compliance and privacy issues, actively engaged citizen scientists to address societal concerns. To ensure ethics, accessibility and diversity, all project materials such as the MOOC training course, advertising, information sheets, ethical application form, participant informed consent form, investigation tools, and surveys were translated from English into a dozen languages, including Greek, Hungarian, Dutch, Finnish, Hebrew, Czech, and Spanish. Coventry University (CU) and its partners secured ethical approval for the project's activities and investigative measures by submitting applications to CU's research ethics panel. This process involved completing CU's online ethics form, addressing a series of research ethics questions, and responding to additional queries from the research ethics panel to gain approval [39]. Throughout the CSI-COP project, three ethics applications were submitted due to a substantial change following the termination of a partner. The study exclusively involved participants aged 18 and above. Participants provided informed consent by physically completing a consent form, signifying their voluntary engagement in the study. They were explicitly informed of the voluntary nature of their participation and their freedom to withdraw at any stage. The consent process was comprehensive, offering clear instructions to participants. They were made aware of their right to withdraw their data from the project until the data destruction date, June 30, 2032, without providing a reason, and this choice would not bear any adverse consequences. Participants were thoroughly briefed about the study's procedures, encompassing the questions they would encounter, the methods of data collection, and the expected duration. The study's confidentiality and data protection measures, adhering to GDPR and the Data Protection Act 2018, were meticulously explained. Personal data underwent anonymization, and stringent security protocols were implemented for both electronic and paper records. To ensure transparency and compliance, we maintained a clear segregation between consent information and research data, bolstering security and mitigating risks related to data breaches. The lead researcher assumes responsibility for data destruction, adhering to the outlined protocols. All data are scheduled for destruction on or before June 30, 2032, marking ten years after the completion of CSI-COP. Participant recruitment commenced on October 1, 2020, and concluded on June 30, 2022.

The CSI-COP project implemented several steps to engage participants and investigate data protection and online privacy issues. These steps were as follows:

1. Recruitment and outreach: Each project partner involved in these tasks reached out to a wide range of organizations and utilized university websites, social media platforms, and the snowball method to attract a diverse pool of potential volunteers.

2. Creation of a MOOC: With its sub-contractor (Privacy Matters) Coventry University created a dedicated MOOC called 'Your Right to Privacy Online' [40]. This was reviewed internally by Stelar, by project partners, and by external parties (UCL, members of the public,

CU's students). This accessible online resource provided practical exercises to explore cookies on websites and apps, helping participants understand the data collected and shared by cookies. In addition, workshops based on the MOOC were organized to provide hands-on learning experiences. The MOOC consisted of five steps, equipping participants with the knowledge and skills to check and block tracking technologies on Android devices.

3. Assessment and certification: Participants were required to answer 10 questions related to privacy aspects and the use of personal data by third parties in online interactions. If participants achieved a score of 8 out of 10, they received a CSI-COP informal education certificate, acknowledging their learning achievements and motivating them to learn more by joining the project as a citizen scientist.

4. Data collection and questionnaires: After the training stage, participants completed an anonymous questionnaire regarding non-identifiable demographics, behavior, satisfaction with the MOOC, and their intentions to become citizen scientists in the project. Interested individuals could request to read the GDPR compliant Participant Information Sheet, and if motivated to join, sign the Informed Consent sheet. This procedure followed CSI-COP project's GDPR compliant data management plans [18].

5. Transition to citizen scientists: Once participants completed the MOOC, they had the opportunity to become volunteer citizen scientists and join the CSI-COP team if signed consent sheet was received by their local CSI-COP partner. CS role involved investigating the extent of data tracking across the Internet in websites and in apps.

6. Data collection and analysis: In the final step, participants collected information on the ease of understanding website privacy policies and cookie notices, using free online privacy audit tools. CS then recorded their findings in a specially created CSI-COP Excel table with relevant data and screenshots.

7. Creation of a Taxonomy of Digital Cookies and Online Trackers: From the investigations of websites and apps, a classification of the different types of cookies and tracking technologies was created. It is accessible from CSI-COP website results page and on open-access platform Zenodo.

8. Innovation of an open-access repository: From the combined efforts of the CSI-COP team and volunteer citizen scientists Coventry University's sub-contractor, Xcel Resources Ltd., innovated a searchable open-access repository. This repository can be explored to find information about the websites and apps investigated in CSI-COP and the number and types of trackers that might be embedded. Stakeholders, anyone who uses the Internet, can explore to determine the presence of trackers on the web pages they might visit.

Through these steps, the CSI-COP project effectively engaged citizen scientists and implemented the principles of CS, including active involvement, meaningful outcomes, mutual benefits, provision of feedback, open access, acknowledgment, and consideration of legal and ethical factors.

## Theoretical framework of the study

As a theoretical framework, this study embraced the ten principles of Citizen Science inclusion established by ECSA [21]. In adhering to these principles, CSI-COP underscores its dedication to upholding best practices in citizen science. This commitment ensures meaningful participant involvement, scientific excellence, and the progressive enrichment of knowledge in the realms of online privacy and citizen science:

1. *Active involvement of citizens in generating new knowledge*: CSI-COP actively engages citizens as collaborators, contributors, and project leaders, granting them significant roles in exploring new knowledge about online privacy and investigating cookies.

2. *Focus on meaningful scientific outcomes with practical applications*: CSI-COP's research on online privacy and cookies aims to address real-world challenges and inform actions such as policy-making and conservation efforts, aligning with the principle of achieving meaningful scientific outcomes with practical applications.

3. *Mutual benefits for professional and citizen scientists*: Both professional and citizen scientists involved in CSI-COP experience mutual advantages, encompassing learning opportunities, personal fulfillment, and contributions to scientific evidence.

4. *Opportunities for citizen participation in multiple stages of research*: CSI-COP enables citizen scientists to engage in various stages of research, spanning from formulating research questions to data analysis and communicating results.

5. *Provision of feedback on data usage and project outcomes*: CSI-COP emphasizes providing feedback to citizen scientists regarding their data usage and the project's outcomes, fostering transparency and accountability.

6. *Recognition of CS as a valuable research approach*: CSI-COP recognizes citizen science as a valuable research approach, treating it on par with traditional research methods and considering its limitations and biases.

7. *Open-access to data and publication of results*: CSI-COP adheres to the principle of open data sharing by creation of an open-access repository and by making project data and meta-data publicly available and publishing results in open access formats wherever possible.

8. *Acknowledgment of CSs in project outcomes*: CSI-COP acknowledges the contributions of CSs in project results and publications, giving them due credit for their involvement.

9. *Evaluation of scientific, data quality, participant experience, and societal impact*: CSI-COP conducts evaluations to assess scientific outputs, data quality, participant experiences, and the broader societal or policy impact, ensuring a holistic understanding of the project's effectiveness.

10. *Consideration of legal, ethical, and environmental factors in project implementation*: CSI-COP integrates legal, ethical, and environmental factors into its implementation, addressing issues like copyright, intellectual property, confidentiality, and environmental impact, thereby ensuring responsible and ethical conduct throughout the project's duration.

This dedication with the ten principles of Citizen Science, ensures that participant engagement is meaningful, scientific standards are rigorously upheld, and knowledge in the fields of online privacy and citizen science continually advances.

**The study data: MOOC participants' surveys.** As mentioned above, the CSI-COP project's team recruited participants, and for this paper we include the eight countries whose citizen scientists had completed the MOOC-based training regarding human rights on the web and completed the anonymous survey. Overall, there were over 600 CSI-COP MOOC completions, and 188 active citizen scientists (as of July, 2023). For the purposes of this study, we refer to the 540 participants, candidate citizen scientists, who had successfully completed the MOOC by the time of a CSI-COP deliverable report [41], and were administered an anonymous survey, which included various demographic questions. Additionally, the survey

collected information on participants' Internet usage practices, satisfaction with the MOOC, and their intentions to join the project as citizen scientists.

It is important to note that participation in the survey was voluntary, and participants had the option to choose "prefer not to say" or refrain from selecting any answer options. These participants were coded as N.A. (not available) participants and are indicated as such in Table 1. Consequently, the number of respondents (N) varies across the different categories presented in Table 1. It is worth mentioning that all surveys were manually filled out in Word files offline. To ensure ethical research practices in the CSI-COP project, the coordinator and partners conducted regular check-ins to ensure compliance with ethical commitments. Table 1 provides an overview of the demographic and background information of the MOOC participants. It includes data on the overall sample, encompassing participants who completed the questionnaire and engaged in the MOOC course but did not become citizen scientists, as well as those who joined the project as citizen scientists.

Among the individuals who have become citizen scientists, the gender distribution is nearly equal. The most prominent age group falls within the range of 18 to 39 years old. Half of the CS were students, with almost 60% being undergraduate students. The largest group of volunteers worked over 36 hours per week and resided in urban areas without any accessibility issues. In terms of language, Greek is the primary language for the majority of the MOOC learners as their mother tongue. Both citizen scientists and non-citizen scientists exhibit a high level of internet access, utilizing various applications on both computers and mobile devices regularly and on a daily basis. A higher proportion of those who became citizen scientists found the MOOC to be highly useful, and a larger percentage of citizen scientists reported studying the MOOC independently (self-learning) compared to those who did not ultimately become citizen scientists. Regarding the "Mother tongue" category, the "other" subcategory includes additional languages with less than four participants each. The languages included in this subcategory are Finnish, Albanian, Russian, Arabic, Catalan, French, Turkish, Bulgarian, Croatian, Cypriot, German, Gujarati, Hindi, Italian, Malayalam, Mandarin, Persian, Polish, Slovakian, Thai, and Urdu. Such a variety of native speaking languages reflects the great cultural and ethnic diversity achieved by the project.

## Statistical data analysis

The research employs a quantitative research. In addition to conducting descriptive analysis on the participants' surveys, we also utilized more advanced statistical techniques to examine the relationships between various research variables. We utilized the Pearson correlation coefficient (PCC) alongside the point-biserial correlation coefficient (PBC) and Kendall's coefficient of rank correlation (tau-sub-b) to evaluate relationships between the variables, following the coding outlined in Table 1. Additionally, we employed different types of tests, such as independent-samples t-tests, one-way ANOVA, and chi-square tests of independence, to examine differences in variance between subgroups. To address the inquiry regarding predictive factors for individuals becoming citizen scientists, we constructed a binary logistic regression model. This model aimed to determine the likelihood of a participant becoming a citizen scientist based on the data provided by the survey. Fig 1 showcases the conceptual research model, where the arrows represent the research questions and the relationships among the constructs.

## Results

### Relationships between peripheral and specific background variables and an individual's status as a citizen scientist

To address the first research question and as a preliminary test and basis for subsequent tests, a Pearson correlation coefficient (PCC) alongside a point-biserial correlation coefficient (PBC)

**Table 1. Socio-demographic and background characteristics, and MOOC-related variables (N = 540).**

| Variable | Values | Code | Overall N = 540 | | Did not become CS N = 370 | | Became CS N = 170 | |
|---|---|---|---|---|---|---|---|---|
| | | | N | 0% | N | 0% | N | 0% |
| Gender (N = 530) (N.A; n = 10, 1.8%) | Male | 0 | 243 | 45.8% | 165 | 45.1% | 78 | 47.6% |
| | Female | 1 | 287 | 54.2% | 201 | 54.9% | 86 | 52.4% |
| Age range (N = 523) (N.A; n = 17, 3.1%) | 18–39 | 1 | 293 | 56.0% | 195 | 53.7% | 98 | 61.3% |
| | 40–65 | 2 | 208 | 39.8% | 153 | 42.1% | 55 | 34.4% |
| | 66+ | 3 | 22 | 4.2% | 15 | 4.1% | 7 | 4.4% |
| Occupation (N = 525) (N.A; n = 15, 2.7%) | Student | 1 | 204 | 38.9% | 134 | 37.0% | 70 | 42.9% |
| | Student and employed | 2 | 72 | 13.7% | 56 | 15.5% | 16 | 9.8% |
| | Employed | 3 | 249 | 47.4% | 172 | 47.5% | 77 | 47.2% |
| Employment hours range (Student and employed + Employed) (N = 318) (N.A; n = 4, 1.2%) | Retired | 1 | 34 | 10.69% | 24 | 10.6% | 10 | 10.9% |
| | Accessibility issues, not able to work | 2 | 2 | 0.63% | 1 | 0.4% | 1 | 1.1% |
| | Not employed, not looking for work | 3 | 8 | 2.52% | 36 | 15.9% | 3 | 3.3% |
| | Not employed, looking for work | 4 | 59 | 18.55% | 5 | 2.2% | 23 | 25.0% |
| | Employed, working 1–36 hrs/w | 5 | 42 | 13.21% | 27 | 11.9% | 15 | 16.3% |
| | Employed, working 36.5+ hrs/w | 6 | 173 | 54.4% | 133 | 58.8% | 40 | 43.5% |
| Student degree (only students) (N = 269) | Undergraduate | 1 | 161 | 59.9% | 114 | 60.6% | 47 | 58.0% |
| | Postgraduate | 2 | 57 | 21.2% | 43 | 22.9% | 14 | 17.3% |
| | Doctoral | 3 | 51 | 19% | 31 | 16.5% | 20 | 24.7% |
| Accessibility issues (N = 519) (N.A; n = 21, 3.8%) | Yes | 0 | 42 | 7.8% | 37 | 10.5% | 5 | 3% |
| | No | 1 | 477 | 88.3% | 315 | 89.5% | 162 | 97% |
| Location (N = 519) (N.A; n = 21, 3.8%) | Urban | 0 | 480 | 92.50% | 331 | 93.8% | 149 | 89.8% |
| | Rural | 1 | 39 | 7.50% | 22 | 6.2% | 17 | 10.2% |
| Mother tongue (N = 503) (N.A; n = 37, 6.8%) | Greek | | 269 | 53.48% | 207 | 55.9% | 62 | 36.5% |
| | Romanian | | 70 | 13.92% | 64 | 17.3% | 6 | 3.5% |
| | Czech | | 51 | 10.14% | 25 | 6.8% | 26 | 15.3% |
| | English | | 28 | 5.57% | 7 | 1.9% | 21 | 12.4% |
| | Hebrew | | 23 | 4.57% | – | – | 23 | 13.5% |
| | Hungarian | | 18 | 3.58% | 10 | 2.7% | 8 | 4.7% |
| | Spanish | | 11 | 2.19% | 10 | 2.7% | 1 | .6% |
| | Other languages (<4 participants from each language) | | 33 | 6.67% | 47 | 12.7% | 23 | 13.5% |
| Internet access (N = 521) (N.A; n = 19, 3.5%) | Have access to own Internet connection | 0 | 498 | 95.60% | 339 | 94.2% | 159 | 98.8% |
| | Use public access | 1 | 23 | 4.40% | 21 | 5.8% | 2 | 1.2% |
| Internet usage (N = 532) (N.A; n = 8, 1.5%) | Less than once a week | 1 | 5 | 0.90% | 3 | .80% | 2 | 1.2% |
| | 2–3 times a week | 2 | 23 | 4.30% | 16 | 4.4% | 7 | 4.2% |
| | Daily | 3 | 504 | 94.70% | 346 | 94.8% | 158 | 94.6% |
| Purpose of Internet usage (N = 524) (N.A; n = 16, 2.9%) | Use the Internet in a limited way | 1 | 3 | 0.60% | 2 | .60% | 1 | .60% |
| | Leisure | 2 | 76 | 14.50% | 51 | 14.0% | 25 | 15.5% |
| | Part of daily work | 3 | 100 | 19.10% | 81 | 22.3% | 19 | 11.8% |
| | Work and leisure | 4 | 345 | 65.80% | 229 | 63.1% | 116 | 72.0% |
| Frequency of apps usage on computer (N = 528) (N.A; n = 12, 2.2%) | Do not use | 1 | 14 | 2.70% | 6 | 1.7% | 8 | 4.8% |
| | Rarely | 2 | 25 | 4.70% | 17 | 4.7% | 8 | 4.8% |
| | Regularly | 3 | 489 | 92.60% | 340 | 93.7% | 149 | 90.3% |

(*Continued*)

**Table 1.** (Continued)

| Variable | Values | Code | Overall N = 540 | | Did not become CS N = 370 | | Became CS N = 170 | |
|---|---|---|---|---|---|---|---|---|
| | | | N | 0% | N | 0% | N | 0% |
| Apps usage: Computers (out of 540) | Work apps | | 375 | 69.40% | 265 | 71.6% | 110 | 64.7% |
| | Entertainments apps | | 275 | 50.90% | 177 | 47.8% | 98 | 57.6% |
| | News apps | | 230 | 42.60% | 152 | 41.1% | 78 | 45.9% |
| | Education apps | | 166 | 30.70% | 105 | 28.4% | 61 | 35.9% |
| | Games apps | | 137 | 35.40% | 84 | 22.7% | 53 | 31.2% |
| | Lifestyle apps | | 120 | 22.20% | 73 | 19.7% | 47 | 27.6% |
| | Other apps | | 53 | 9.80% | 26 | 7.0% | 27 | 15.9% |
| Frequency of apps usage on mobile (N = 513) (N.A; n = 27, 5.0%) | Do not use | 1 | 7 | 1.40% | 7 | 2.0% | – | – |
| | Rarely | 2 | 41 | 8.00% | 32 | 9.1% | 9 | 5.6% |
| | Regularly | 3 | 465 | 90.60% | 314 | 89.0% | 151 | 94.4% |
| Apps usage: Mobile (out of 540) | News apps | | 233 | 43.10% | 139 | 37.6% | 94 | 55.3% |
| | Entertainments apps | | 233 | 43.10% | 150 | 40.5% | 83 | 48.8% |
| | Education apps | | 190 | 35.20% | 128 | 34.6% | 62 | 36.5% |
| | Lifestyle apps | | 170 | 31.50% | 102 | 27.6% | 68 | 40.0% |
| | Games apps | | 168 | 31.10% | 104 | 28.1% | 64 | 37.6% |
| | Other apps | | 171 | 31.70% | 98 | 26.5% | 73 | 42.9% |
| Satisfaction from the MOOC (N = 356) (N.A; n = 184, 34%) | Not useful | 1 | 4 | 1.10% | 2 | 9.0% | 2 | 1.6% |
| | No thoughts | 2 | 9 | 2.50% | 5 | 2.1% | 4 | 3.3% |
| | Useful | 3 | 100 | 28.10% | 74 | 31.6% | 26 | 21.3% |
| | Very useful | 4 | 243 | 68.30% | 153 | 65.4% | 90 | 73.8% |
| Mode of course completion (N = 540) | Self-learning | 1 | 163 | 30.20% | 91 | 24.6% | 72 | 42.4% |
| | Online workshop (by team partner member) | 2 | 57 | 10.60% | 51 | 13.8% | 6 | 3.5% |
| | Face-to-face workshop (by team partner member) | 3 | 320 | 59.30% | 228 | 61.6% | 92 | 54.1% |

*Notes*: a) N.A. denotes the number of participants who chose the answer option 'prefer not to say' or did not choose any of the answer options. b) Employed are individuals who are actively working and earning income through gainful employment.

and Kendall's coefficient of rank correlation tau-sub-b were calculated. These calculations aimed to determine the relationships between the *socio-demographic* variables, the *frequency of apps usage on computers/mobile* with the *satisfaction with MOOC*, the *behavioral intentions to become a citizen scientist* and *becoming a citizen scientist*. Table 2 presents the tests results.

As can be observed from Table 2, the intention to join the CSI-COP project showed a positive and weak correlation with gender, indicating that females had a greater tendency to express interest in joining the project, and they were more satisfied than the males in relation to the MOOC. However, upon further analysis, it was found that there was no significant relationship between gender and actual participation as citizen scientists in the project. Next, the intention to become a citizen scientist in the CSI-COP project exhibited a negative correlation with accessibility issues. Participants facing accessibility challenges displayed a higher intention to participate actively in the project as citizen scientists. Although in practice, participants with accessibility problems tended to have significantly lower levels of participation as citizen scientists compared to participants without such issues. In addition, those who reported high levels of satisfaction with the MOOC, were older, more educated and had higher number of apps on both mobile devices and computers. Furthermore, there was a positive correlation

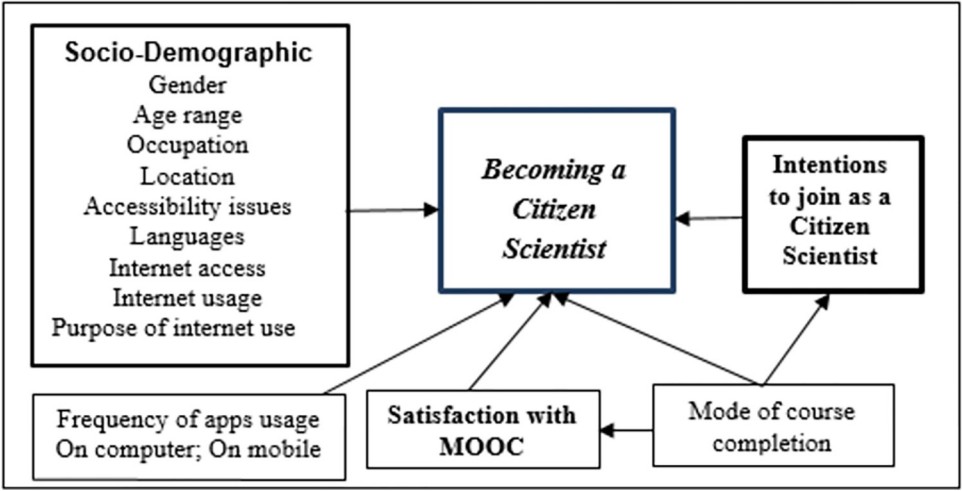

**Fig 1. Research model.**

- In bold: dependent variables; in bold and italic: the main dependent variable in the study.

- Socio-demographic characteristics' variables, mode of course completion, frequency of apps usage on computer and on mobile as independent variables

.

between satisfaction with the MOOC and the intention to join CSI-COP, indicating that those who were more satisfied with the MOOC were more likely to express an intention to participate in the project. Finally, becoming a citizen scientist in the CSI-COP project was positively correlated to internet access, the number of apps installed on both mobile devices and computers, and the frequency of apps usage on mobile. Additionally, the intention to join and become a citizen scientist displayed positive correlations with actually becoming a citizen scientist in practice. This indicates that participants who expressed a stronger intention to become citizen scientists were more likely to follow through and become actively involved in the project.

**The intention to become a citizen scientist vs. actually becoming a citizen scientist.** Among the 540 survey participants considered for this paper, 96.4% rated the MOOC as useful and very useful. However, not all MOOC completions translated into interest in joining the CSI-COP project as citizen scientists. Out of the 540 MOOC completions, 170 participants (31.5%) completed both the MOOC and the survey and joined the project team as citizen scientists at the time of writing. Meanwhile, 370 of the 540 participants (68.5%) did not become citizen scientists. Of the 540 participants, 496 individuals provided responses about their intention to participate in the CSI-COP project as citizen scientists (44 participants preferred not to answer). The responses were categorized into four levels in ascending order as follows:

1. A total of 89 participants (17.94%) responded negatively ("no"), indicating that they do not intend to join.

2. 89 participants (17.94%) stated that they required additional information before making a decision.

3. A larger group of 171 participants (34.48%) expressed their potential interest in joining the project.

**Table 2. Correlations of the study's variables, the behavioral intentions to become a CS and actually becoming a citizen scientist.**

| | Gender | Age | Location | Accessibility issues | Occupation | Student Degree | Internet access | Internet usage | Purpose of Internet use | Apps usage on computer | Apps no. on computer | Apps usage on mobile | Apps no. on Mobile | MOOC Satisfaction | Intention to join CSI-COP |
|---|---|---|---|---|---|---|---|---|---|---|---|---|---|---|---|
| **Satisfaction from the MOOC** | .106* | .271** | -.064 | -.003 | .165** | .257** | .030 | .025 | -.068 | -.059 | -.132* | -.071 | -.139* | – | |
| **Intention to join CSI-COP** | .119** | .033 | .063 | -.117* | -.059 | .107 | -.052 | .06 | -.009 | .037 | -.044 | .064 | -.024 | -.195** | – |
| **Became a citizen scientist** | -.023 | -.058 | .071 | .129** | -.031 | .063 | -.103* | -.009 | .045 | -.08 | .179** | .098* | .215** | .046 | .172** |

*Notes*: 1) The coding of the variables is shown in Table 1. 2) According to Cohen's recommended guidelines [42]: |r| < 0.3 → Weak relationship

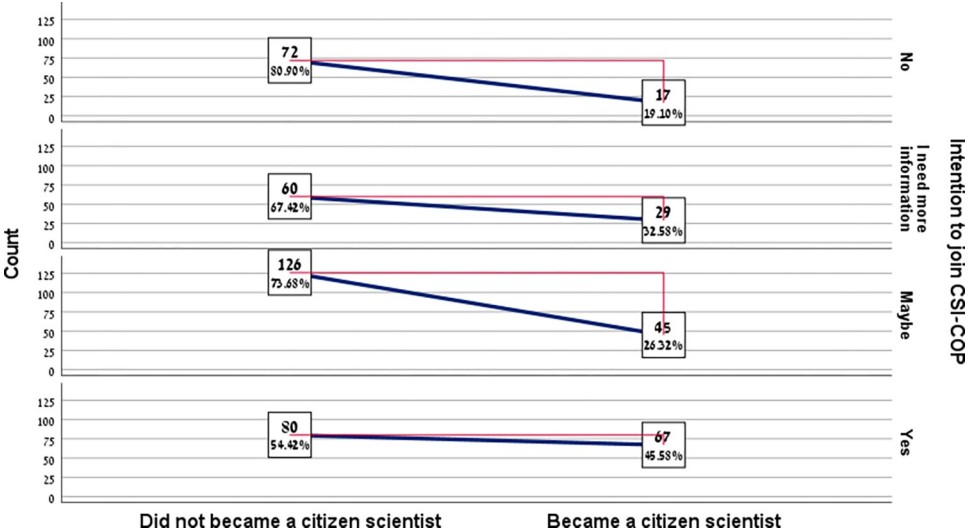

**Fig 2. The proportion of participants who actually became citizen scientists at each intention to join level (n = 496).** *Note*: The red line represents the gap between the two groups at each level of the intentions to join the project.

4. Finally, 147 participants (29.64%) confirmed their intention to join the project as citizen scientists by responding positively with a "yes".

According to the second research question, we examined the effect of individuals' intention to participate in the project as citizen scientists on their actual engagement and status as citizen scientists in practice. Fig 2 presents the proportion of participants who *became citizen scientists* at each intention to join level:

An independent-samples t-test was performed to determine whether there is a significant difference between the participants who became citizen scientists and those who did not become citizen scientists regarding their level of intention to participate in the project (a continuous variable). The results of the analysis indicated a significant difference, $t(494) = -3.869$, $p < .001$. The intention to join among the participants who became citizen scientists was found to be significantly higher ($M = 3.03$, $SD = 1.02$) compared to those who did not become citizen scientists ($M = 2.63$, $SD = 1.06$). *Cohen's d*, resulting in a value of 0.37, which is considered a small to medium effect.

**The effect of the languages.** The third research question explored the impact of variances in participants' native languages on the research variables. Table 3 illustrates the distribution of categorical variables, while Table 4 presents descriptive statistics and test results for quantitative continuous variables across the eight participating countries in the CSI-COP project. Following Table 3, the disparities were assessed using the chi-square ($X^2$) goodness of fit test, comparing observed proportions among countries for each research variable to determine their significance.

As can be observed from Table 3, the largest group is the Greek-speaking group while the smallest is the Spanish-speaking group. However, only 23% of the Greek speakers became actual citizen scientists compared to 75% of the English speakers and 100% of the Hebrew speakers. Among the Romanian speakers, which was the second largest group, only 9% became citizen scientists in the project. These differences were found to be significant ($X^2(7) = 113.55$, $p < .001$). Participants in different languages also significantly differed in the manner of the course ($X^2(14) = 645.47$, $p < .001$). Of the Greek participants, 95% attended the face-to-face

**Table 3. Distribution of participants according to their languages.**

| Variable | Value | Mother tongue | | | | | | | | | | | | |
|---|---|---|---|---|---|---|---|---|---|---|---|---|---|---|
| | | Greek (N = 269) | | Romanian (N = 70) | | Czech (N = 51) | | English (N = 28) | | Hebrew (N = 23) | | Hungarian (N = 18) | | Spanish (N = 11) | |
| | | N | % | N | % | N | % | N | % | N | % | N | % | N | % |
| **Become a citizen scientist** | Did not become a citizen scientist | 207 | **77%** | 64 | 91% | 25 | 49% | 7 | 25% | 0 | 0% | 10 | 56% | 10 | 91% |
| | Became a citizen scientist | 62 | 23% | 6 | 9% | 26 | 51% | 21 | **75%** | 23 | **100%** | 8 | 44% | 1 | 9% |
| **Mode of MOOC completion** | Self-learning | 13 | 5% | 18 | 26% | 51 | **100%** | 21 | **75%** | 12 | 52% | 4 | 22% | 11 | **100%** |
| | Online workshop | 0 | 0% | 52 | 74% | 0 | 0% | 1 | 4% | 0 | 0% | 0 | 0% | 0 | 0% |
| | Face-to-face workshop | 256 | **95%** | 0 | 0% | 0 | 0% | 6 | 21% | 11 | 48% | 14 | **78%** | 0 | 0% |
| **Gender** | Male | 122 | 46% | 18 | 26% | 24 | 47% | 16 | **59%** | 11 | 48% | 7 | 41% | 9 | 82% |
| | Female | 141 | 54% | 52 | **74%** | 27 | 53% | 11 | 41% | 12 | 52% | 10 | **59%** | 2 | 18% |
| **Age range** | 18–39 | 121 | 45% | 32 | 46% | 34 | **68%** | 20 | **91%** | 19 | **86%** | 12 | 67% | 6 | 55% |
| | 40–65 | 140 | 53% | 37 | 54% | 3 | 6% | 2 | 9% | 3 | 14% | 5 | 28% | 5 | 45% |
| | 66+ | 5 | 2% | 0 | 0% | 13 | 26% | 0 | 0% | 0 | 0% | 1 | 6% | 0 | 0% |
| **Location** | Urban | 241 | 94% | 66 | 94% | 41 | 84% | 27 | 96% | 16 | 70% | 16 | 100% | 9 | 90% |
| | Rural | 15 | 6% | 4 | 6% | 8 | 16% | 1 | 4% | 7 | 30% | 0 | 0% | 1 | 10% |
| **Occupation** | Student | 74 | 28% | 33 | 47% | 31 | **61%** | 13 | **48%** | 10 | 45% | 6 | 33% | 2 | 18% |
| | Student and employed | 43 | 16% | 6 | 9% | 7 | 14% | 2 | 7% | 0 | 0% | 2 | 11% | 3 | 27% |
| | Employed | 145 | 55% | 31 | 44% | 13 | 25% | 12 | 44% | 12 | 55% | 10 | 56% | 6 | 55% |
| **Student Degree** | Undergraduate | 88 | **77%** | 16 | 41% | 20 | 53% | 1 | 10% | 10 | **100%** | 2 | 25% | 1 | 20% |
| | Postgraduate | 22 | 19% | 4 | 10% | 16 | 42% | 1 | 10% | 0 | 0% | 2 | 25% | 3 | 60% |
| | Doctoral | 5 | 4% | 19 | 49% | 2 | 5% | 8 | 80% | 0 | 0% | 4 | 50% | 1 | 20% |
| **Internet access** | Have access to own connection | 242 | 94% | 70 | 100% | 50 | 98% | 28 | 100% | 23 | 100% | 18 | 100% | 11 | 100% |
| | Use public access | 16 | 6% | 0 | 0% | 1 | 2% | 0 | 0% | 0 | 0% | 0 | 0% | 0 | 0% |
| **Accessibility issues** | Yes | 11 | 4% | 21 | **31%** | 1 | 2% | 1 | 4% | 1 | 5% | 0 | 0% | 0 | 0% |
| | No | 250 | 96% | 47 | 69% | 49 | 98% | 25 | 96% | 21 | 95% | 18 | 100% | 10 | 100% |

**Table 4. The effect of the different languages on the study variables.**

| | | A | B | C | D | E | F | G | F test |
|---|---|---|---|---|---|---|---|---|---|
| | | Greek (N = 269) | Romanian (N = 70) | Czech (N = 51) | English (N = 28) | Hebrew (N = 23) | Hungarian (N = 18) | Spanish (N = 11) | |
| **Intention to join CSI-COP** | M | 2.64 | 3.33 | 2.18 | 3.11 | **3.70** | 2.61 | 3.00 | $F(7,488) = 9.77, p < .001$ |
| | SD | 1.01 | 0.93 | 0.95 | 1.09 | 0.7 | 0.98 | 0.82 | E>A,C,F; B>A,C |
| **Satisfaction from the MOOC** | M | 3.67 | 3.86 | 3.00 | 3.64 | 3.39 | 3.69 | 4.00 | $F(7,348) = 1.87, p > .05$ |
| | SD | 0.57 | 0.36 | 0 | 0.49 | 0.72 | 0.48 | 0 | |
| **Internet usage** | M | 2.91 | 3.00 | 2.94 | 3 | 2.96 | 3 | 3 | $F(7,524) = 1.32, p > .05$ |
| | SD | 0.33 | 0.0 | 0.24 | 0 | 0.21 | 0 | 0 | |
| **Frequency of apps usage on computer** | M | 2.89 | 2.91 | 2.94 | 2.86 | 2.91 | 2.94 | 2.82 | $F(7,520) = .350, p > .05$ |
| | SD | 0.42 | 0.28 | 0.31 | 0.45 | 0.29 | 0.24 | 0.6 | |
| **Apps number (on computer)** | M | 2.83 | **1.44** | 3.60 | 2.92 | **4.19** | 3.06 | 3.90 | $F(7,476) = 14.36, p < .001$. E>A,B; B<A,C,D,E,F,G |
| | SD | 1.5 | 0.99 | 1.38 | 1.1 | 1.94 | 1.56 | 1.1 | |
| **Frequency of apps usage on mobile** | M | 2.89 | 2.87 | 2.92 | 3.00 | 2.96 | 2.81 | 2.91 | $F(7,505) = .84, p > .05$ |
| | SD | 0.36 | 0.38 | 0.34 | 0 | 0.21 | 0.54 | 0.3 | |
| **Apps number (on mobile)** | M | 2.48 | **1.37** | 3.21 | 2.96 | **4.27** | 2.50 | 2.78 | $F(7,450) = 12.72, p < .001$. E>A,B,D,F; B<A,C,D,E |
| | SD | 1.43 | 0.84 | 1.68 | 1.34 | 1.45 | 1.4 | 1.09 | |

workshop, as did 78% of the Hungarian speakers. In contrast, 74% of the Romanian speakers took an online workshop by an instructor. Among the English-speaking participants, 75% preferred to study the MOOC independently (self-learning). Significant differences were found between the speakers of the languages also in relation to gender ($X^2(7) = 20.74$, $p < .05$), age ($X^2(14) = 130.84$, $p < .001$), location (although most of the participants came from an urban location) ($X^2(7) = 27.10$, $p < .001$), occupation ($X^2(14) = 45.54$, $p < .001$) and student degree ($X^2(14) = 96.95$, $p < .001$). Regarding the accessibility issues ($X^2(7) = 59.42$, $p < .001$), among participants who spoke all the languages, more than 90% indicated that they did not encounter any accessibility issues. However, it is noteworthy that among Romanian speakers, 31% of the participants reported experiencing accessibility problems. In relation to Internet access, most participants, regardless of their language and without significant differences, use their own access to the Internet ($X^2(7) = 13.67$, $p > .05$).

Next, Table 4 presents the means, standard deviations, and results of the ANOVA tests conducted on the continuous research variables across different language groups. The findings reveal a significant main effect on the intentions to participate in the CSI-COP project, as well as the number of applications installed on both computers and mobile devices. To identify specific significant differences, post hoc comparisons were conducted using the Bonferroni correction (Table 4).

According to the pairwise comparisons, Hebrew speaking participants reported intentions to join as citizen scientists significantly more than Greek, Czech and Hungarian speakers. Romanian speakers reported higher intentions than Greek and Czech speakers, while in practice, only 9% of the participants who speak the Romanian language, joined the project as citizen scientists (see Table 3). In relation to the number of apps installed on mobile phones and computers, on average, Hebrew speakers reported the highest number of installed apps, whereas Romanian speakers reported the lowest number.

## The effect of the mode of the course completion

To answer the fourth research question, we examined whether there are differences between the *mode of the course completion* (self-learning, online workshop, face-to-face workshop) and the level of satisfaction from the MOOC, the participant intention to join CSI-COP as a citizen scientist and becoming a citizen scientist in practice. First, a one-way ANOVA was conducted to determine the effect of the mode of the course completion on the satisfaction from the MOOC. The results indicate a non-significant effect, ($F(2, 353) = 1.690$, $p = .186$). There were no significant differences between participants who self-studied the MOOC (M = 3.54, SD = .66) and those who participated in online workshops (M = 4.00, SD = .03) or in face-to-face workshops organized by team partner members (M = 3.66, SD = .57) in their level of satisfaction with the course. Subsequently, a one-way ANOVA was conducted to determine the effect of the mode of the course completion on the participant's intention to join CSI-COP as a citizen scientist. The results indicate a significant effect, ($F(2, 493) = 19.360$, $p < .001$). Post hoc analyses using the Scheffé post hoc criterion for significance indicated that those who participated in the online workshops (M = 3.54, SD = .78) reported significantly higher intentions to be citizen scientists ($p < .001$) than those who self-learnt the MOOC (M = 2.73, SD = 1.12) or those who attended face-to-face workshops (M = 2.62, SD = 1.02). The pairwise comparison of the MOOC individual participation group with the face-to-face attendance group was non-significant. Next, a chi-square test of independence was performed to examine the relation between the mode of the course completion and becoming a citizen scientist in practice. The relation between these variables was significant, $X^2(2, N = 370) = 24.88$, $p < .001$. Participants who attended the face-to-face workshops were more likely to not become a citizen scientist

**Table 5. Cross-tabulation of mode of course completion and becoming a citizen scientist.**

| Mode of course completion | Did not become a citizen scientist | | Become a citizen scientist | | Total | |
|---|---|---|---|---|---|---|
| | N | % | N | % | N | % |
| Self-learning | 91 | 24.60% | 72 | 42.40% | 163 | 30.20% |
| Online workshop | 51 | 13.80% | 6 | 3.50% | 57 | 10.60% |
| Face-to-face workshop | 228 | 61.60% | 92 | 54.10% | 320 | 59.30% |
| Total | **370** | 100.00% | **170** | 100.00% | 540 | 100.00% |

than those who participated online or via self-learning. Table 5 presents the test results in cross tabulation of the two variables.

As evident from the data analysis, the face-to-face group comprised the largest number of participants (N = 320). However, the majority of participants who did not become citizen scientists belonged to this group, accounting for the largest proportion (61.60%).

## Prediction model for becoming a citizen scientist

A binary logistic regression was performed to ascertain the effects of the study variables on the likelihood that participants become citizen scientists. To assess potential multicollinearity among the independent variables, the Variance Inflation Factor (VIF) technique was employed. VIF values, ranging from 1.083 to 2.037, comfortably remained below the critical threshold of 5. These values indicate moderate correlations, affirming the robustness and reliability of the regression model. The logistic regression model was statistically significant, $\chi^2(14) = 29.639$, $p = .009$. The model explained 33.0% (Nagelkerke $R^2$) of the variance in becoming a citizen scientist, and correctly classified 71.5% of cases. Table 6 presents the test results.

According to the regression results, among the demographic variables and the specific MOOC related variables, undergraduate students were 6.84 times more likely to become citizen scientists than postgraduate and doctoral students. As could be expected, increasing the

**Table 6. Logistic regression model for becoming a citizen scientist.**

| | B | S.E. | Exp(B) | Wald | Sig. |
|---|---|---|---|---|---|
| Gender (1) | -.056 | .527 | 0946 | .011 | .916 |
| Age | -1.352 | .818 | .259 | 2.732 | .098 |
| Location(1) | .741 | .908 | 2.099 | .667 | .414 |
| Accessibility issues(1) | 2.661 | 1.587 | 14.308 | 2.811 | .094 |
| Occupation(1) | .333 | 0.744 | 1.395 | .201 | .655 |
| Undergraduate student | | | | 6.836 | **.033** |
| Postgraduate student | -2.012 | .804 | .134 | 6.269 | **.012** |
| Doctoral student | -2.382 | 1.045 | .092 | 5.196 | **.023** |
| Internet access(1) | -.089 | 1.283 | .915 | .005 | .945 |
| Apps number (Computer) | -.378 | .247 | .685 | 2.341 | .126 |
| Apps number (Mobile) | .391 | .233 | 1.479 | 2.818 | .093 |
| Self-learning the MOOC | | | | .555 | .758 |
| Online workshop | -.486 | .652 | .615 | .555 | .456 |
| Face-to-face workshop | -2.389 | 276.889 | .002 | .001 | .999 |
| Satisfaction from the MOOC | -.079 | .367 | .924 | .047 | .828 |
| Intention to join CSI-COP | .459 | .234 | 1.582 | 3.85 | **.049** |

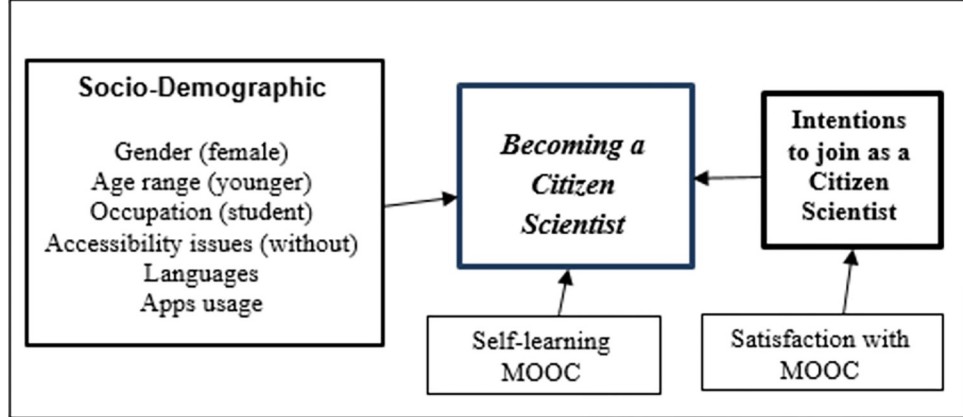

**Fig 3. The empirical research model.**

*intention to join CSI-COP* project was associated with an increased likelihood of becoming citizen scientist in practice. In addition, the analysis revealed that the younger age of the participants, a larger number of mobile applications, and the absence of accessibility problems were found to be marginally significant predictors of actual participation as citizen scientists. While not reaching full significance, these factors showed a notable trend towards influencing participants' engagement in the project as citizen scientists.

## The empirical research model

Fig 1 in this article depicted the conceptual theoretical research model, including research questions and relationships among constructs. Fig 3 summarizes the main results of the study into the comprehensive empirical model for inclusive citizen science.

As can be viewed from Fig 3, there are strong relationships between gender, age, occupation, accessibility, languages/cultures, and app usage and becoming a citizen scientist. Specifically, women, young participants, students, participants without accessibility problems, and those who extensively used applications on computers and mobile phones were more likely to become actual citizen scientists. Additionally, higher satisfaction levels with the MOOC were associated with a stronger intention to join, while self-learners showed the highest proportion of individuals becoming citizen scientists. Importantly, the study highlighted the significance of motivation, as the participant's behavioral intentions strongly predicted their actual participation in the project. The study also revealed that the participant's language (i.e. ethnicity and culture) had an impact on their decision to join the project, indicating that individuals from different language backgrounds may have varied inclinations or motivations when it comes to participating in the citizen science initiative.

## Discussion

The objective of the present study was to identify the characteristic traits of active citizen scientists that may help increase inclusive engagement in CS projects. The study's methodology aligns with the goal of inclusivity in citizen science by leveraging and applying the ten principles of citizen science inclusion formulated by ECSA. This paper focuses on the 540 participants who successfully completed CSI-COP's MOOC as reported in the project's deliverable report (D4.3|D17, https://csi-cop.eu/project-results/who-are-citizen-scientists/) about *who are CSI-COP's citizen scientists* [42]. The MOOC created by Coventry University and its sub-

contractor (Privacy Matters) included an anonymous survey, which encompassed a range of demographic and behavioral questions, including gender, age range, geographical location, and socio-economic status. Among the 540 volunteer participants who filled out the surveys, 170 joined the project team as citizen scientists and contributed to the investigation of cookie notices and privacy policies to ensure transparency and informed consent regarding tracking activities by first or third parties. The study included 243 (46%) male and 287 (55%) female participants, with the gender distribution remaining consistent between those who became citizen scientists and those who did not. Additionally, the majority of the participants fell within the age range of 18 to 39 years old. Approximately half of the citizen scientists were students, with nearly 60% of the students being undergraduate students. The largest group of volunteers worked over 36 hours per week and resided in urban areas without any accessibility issues. In terms of language, the study identified seven dominant language groups, each consisting of over 10 participants: Greek, English, Hebrew, Czech, Hungarian, Romanian, and Spanish.

The findings from this study shed light on several key relationships and factors related to the intention and actual participation of individuals as citizen scientists in the CSI-COP project. Firstly, the analysis revealed a positive correlation between *gender* and the intention to join the project, suggesting that females showed a greater interest in becoming citizen scientists, a finding that reinforces previous studies [5, 16]. Secondly, participants facing *accessibility* challenges expressed a higher intention to actively participate as citizen scientists, but in practice, their levels of participation were lower compared to participants without such issues. This highlights the need for targeted strategies to address accessibility barriers and ensure greater inclusion of individuals facing such challenges [34]. Thirdly, participants with a higher level of satisfaction with the MOOC were more likely to express an intention to join the project as citizen scientists. Moreover, these participants (who were more satisfied with the MOOC) were typically older, more educated, and had a higher number of apps installed on their mobile devices and computers. This suggests that positive experiences with the MOOC and higher technological proficiency were associated with a stronger inclination to participate in the project.

As suggested in the literature, to enhance engagement, CS project should prioritize creating a positive and inclusive experience, fostering knowledge sharing and enthusiasm [27]. Furthermore, the findings revealed positive correlations between becoming a citizen scientist and factors such as *Internet access*, the number of *apps installed* on both mobile devices and computers, and the frequency of *app usage* on mobile devices. This implies, that individuals with greater access to technology and higher levels of engagement with apps were more likely to become actively involved in a CS project as was also reported in previous studies [4, 12]. Lastly, there was a positive relationship between the *intention to join* the project as a citizen scientist, and the act of becoming a citizen scientist in practice. This suggests that individuals who expressed a strong intention to participate were more likely to follow through and actively engage in the project. These findings underscore the complexity of factors influencing the intention and actual participation of individuals as citizen scientists. They emphasize the importance of addressing demographic variables such as accessibility challenges, providing positive learning experiences, and ensuring adequate technological resources to foster greater engagement and participation in citizen science initiatives.

Moreover, by investigating the link between behavioral intention and the act of becoming a citizen scientist, we can enhance our understanding of the underlying motivations and mechanisms that drive individuals to participate in citizen science initiatives. The concept of behavioral intention, introduced by Fishbein and Ajzen [43], is relevant in understanding and predicting certain behaviors. In their theory of reasoned action (TRA), Fishbein and Ajzen proposed that behavioral intentions serve as a predictor of actual behavior. Therefore, the

second research question focused on examining the relationship between individuals' intention to participate as citizen scientists in the CSI-COP project and their actual engagement and status as citizen scientists in practice.

The analysis of the surveys conducted in this study revealed that the intention to join was found to be higher among those who ultimately became citizen scientists compared to those who did not participate as citizen scientists. On the other hand, we also found that about 74% of the survey participants who reported that they "maybe" will become citizen scientists chose not to do this in the end. Interestingly, 20% of those who did not intend to become citizen scientists at all according to the surveys, changed their mind and did join the project. These findings suggest that there is a notable gap between individuals' intention to participate and their actual engagement as citizen scientists in the CSI-COP project. While a substantial number of participants completed the MOOC, only less than a third of them translated their intention into active involvement. The literature indicates that there are various factors identified as influencing behavioral intentions, which, in turn, have a direct or indirect impact on actual behavior [44, 45]. Thus, this highlights the need for further exploration of the factors influencing the transition from intention to actual engagement and the development of targeted strategies to enhance recruitment and retention of citizen scientists.

The inclusion of the MOOC and the specialized training sets our project apart, making it more demanding in terms of participant involvement. The need for participants to acquire specific expertise and skills in online privacy, GDPR and understanding of judicial texts adds an extra layer of complexity to the project. Another challenge stems from that fact that this is the first project, to the best of our knowledge, that involves citizen scientists in investigating and evaluating societal, ethical, judicial, and technical aspects of online information systems, which requires an inter-disciplinary view and analysis, and consequently, much higher digital literacy and cognitive effort from citizen scientists than in a typical CS project. Studies suggest that several factors, including the utilization of technology, aligning the project's language with that of the participants, and incorporating relevant knowledge that is beneficial to the participants, facilitate their engagement in citizen science initiatives [4, 14, 23].

With respect to the fourth research question, we found that participants who completed the MOOC in the self-learning manner, those who participated in online or face-to-face workshops organized by team partners showed the same level of satisfaction with the MOOC. However, participants who took part in online workshops reported on *higher intentions to become citizen scientists* compared to those who learnt the MOOC on their own or participated in face-to-face workshops. Furthermore, the mode of the MOOC completion was found to be related to participants' actual engagement as citizen scientists. Specifically, participants who attended face-to-face workshops were more likely to *not to become citizen scientists* compared to those who participated in online workshops or via self-learning. Although the face-to-face workshop participant group comprised the largest number of participants (N = 320), the majority of participants who did not become citizen scientists belonged to this group, representing the largest proportion (61.60%). These results emphasize the importance of considering different modes of course completion in citizen science projects and their potential influence on participants' commitment and involvement. The inclusion of the MOOC demands a significant investment of time, knowledge, and commitment from participants. Thus, overcoming these challenges and successfully recruiting participants who are willing to invest the required time and effort has been a significant undertaking for our project [8, 13].

Another distinctive element of our project is its international nature, involving participants from various countries, ethnicities, and cultures. This characteristic presents an opportunity to explore the distribution of demographic variables among participants based on their origin, as indicated by their mother tongue. Additionally, it is crucial to investigate the impact of

participants' culture on their intention to actively join and participate as citizen scientists in the project [4]. We identified seven major languages (mother tongue) groups, each consisting of over 10 participants at the time of the Rigler et al [41] report: Greek (269), Romanian (70), Czech (51), English (28), Hebrew (23), Hungarian (18), and Spanish (11) (These numbers have now all increased). In addition to these seven language groups, there were also 33 participants who spoke other languages, with fewer than four participants from each language. These languages included Finnish, Albanian, Russian, Arabic, and more. According to the results, the Greek-speaking participants constituted the largest group, while the Spanish-speaking group was the smallest. However, only 23% of Greek speakers became actual citizen scientists, compared to 75% of English speakers and 100% of Hebrew speakers. Among Romanian speakers, the second largest group, only 9% became citizen scientists in the project. These differences were found to be statistically significant. Participants from different cultures also exhibited significant variations in their engagement with the MOOC. The majority of Greek participants (95%) and Hungarian speakers (78%) preferred attending face-to-face workshops, whereas 74% of Romanian speakers favored an online workshop with an instructor. Among English-speaking participants, 75% opted for self-study using the MOOC.

Significant differences were observed between language groups in terms of gender, location (with the majority of participants from urban areas), occupation, and education. Regarding accessibility issues, the majority of participants across all languages (over 90%) reported no problems. However, it is worth noting that among Romanian speakers, 31% reported experiencing accessibility issues. In terms of Internet access, the vast majority of participants, regardless of language, had their own Internet access, without significant differences. When considering the number of apps installed on mobile phones and computers, Hebrew speakers reported the highest average number of installed apps, while Romanian speakers reported the lowest. The results indicated that participants who spoke Hebrew had significantly higher intentions to join as citizen scientists compared to those who spoke Greek, Czech, and Hungarian. Moreover, in practice, all Hebrew-speaking participants became citizen scientists, demonstrating a 100% participation rate. Romanian speakers reported higher intentions than Greek and Czech speakers. However, in practice, only 9% of Romanian-speaking MOOC learners actually joined the project as citizen scientists. One possible explanation to these findings may be that in some cultures the completion of the training phase and following surveys is motivated by the willingness to actually participate and contribute to the CS project, while in other cultures citizens are more easily engaged in training and learning, as they see gaining knowledge as a more important and motivating goal rather than the participation in the concrete CS project using this knowledge.

These findings highlight the possible influence of participants' cultures and ethnicities on their intentions to join as citizen scientists, their actual engagement in the project, preferred learning methods, and other socio-demographic characteristics. The results underscore the importance of considering culture-specific factors and tailoring engagement strategies to different cultural and ethnic groups to promote inclusivity and maximize participation in future citizen science projects [26, 30].

In the final stage of the analysis, a regression model was conducted to examine the influence of various study variables, including participants' demographics and MOOC-related variables, on their likelihood of becoming citizen scientists. The purpose of this analysis was to identify significant factors that could help explain and predict participants' engagement in the project as citizen scientists. The regression results revealed several significant predictors of participants becoming citizen scientists. First, as expected, the *intention to join the CSI-COP project* was identified as a significant predictor of becoming citizen scientists in practice. Participants who expressed a stronger intention to join the project were more likely to become citizen

scientists in practice. This finding aligns with expectations and highlights the importance of fostering and nurturing participants' interest and motivation to increase their engagement in citizen science initiatives. Another significant predictor identified in the analysis was being a student. We found that undergraduate students stood out as being more likely to become citizen scientists compared to postgraduate and doctoral students. Findings from studies support the notion that the majority of participants in citizen science activities tend to hold an academic education [46]. This finding suggests that students, especially those individuals pursuing undergraduate degrees were more actively involved in the project, potentially due to factors such as their availability, motivation, or alignment with the project's objectives [5, 16].

In addition to these significant predictors, the analysis also revealed some variables that showed a notable trend towards influencing participants' likelihood of becoming citizen scientists. These variables included younger age, a larger number of mobile applications, and the absence of accessibility problems. Younger age was associated with a higher likelihood of active engagement, indicating that younger individuals may possess characteristics or attributes that align well with the project's demands and requirements. The finding suggests in addition that individuals who actively utilize and engage with mobile applications may be more inclined to participate in citizen science projects. Additionally, the absence of accessibility problems, although marginally significant, indicated that providing accessible platforms and resources can positively impact participants' actual engagement as citizen scientists. It highlights the need for inclusive design and accommodations to ensure equal opportunities for individuals with diverse accessibility needs [28, 47].

## Conclusions and implications

This study provides valuable insights into the factors that influence individuals' intention and actual participation as citizen scientists in the CSI-COP project. The findings demonstrate the complex interplay of personal, socio-demographic, and background characteristics, as well as MOOC-related variables, in shaping individuals' engagement in citizen science initiatives. The study aligns with the goal of inclusivity in citizen science by leveraging and applying the ten principles of citizen science inclusion formulated by ECSA [22]. The analysis revealed that gender, accessibility challenges, high satisfaction with the training program, technological proficiency, and factors related to Internet access and app usage significantly influenced participants' intention and actual involvement as citizen scientists. Moreover, the study identified the importance of addressing demographic variables, such as accessibility challenges, and providing positive learning experiences and adequate technological resources to foster greater engagement and participation in citizen science projects.

The findings of this study, in alignment with the principles of citizen science inclusion, have several implications for future citizen science projects and initiatives:

1. *Targeted recruitment strategies*: Inclusivity should be a core consideration in recruitment strategies. The study highlights the need for targeted recruitment strategies to encourage participation among underrepresented groups. Efforts should focus on engaging individuals who face accessibility challenges, as they expressed a higher intention to participate but exhibited lower levels of actual involvement. By addressing accessibility barriers and providing inclusive design, citizen science projects can ensure greater inclusion and participation from individuals with diverse needs.

2. *Promoting positive learning experiences*: Inclusivity requires creating positive and inclusive learning experiences. The study underscores the importance of creating positive learning experiences through MOOCs and training programs. Participants with high levels of

satisfaction with the MOOC were more likely to express an intention to join the project. Since previous research found a gender gap in online privacy literacy [48], projects must provide enhanced training programs to women to eliminate this gap. Projects should design educational materials that are accessible, culturally sensitive, and linguistically diverse to cater to a broad range of participants. By designing high-quality, engaging, and informative educational materials, citizen science initiatives can enhance participants' motivation and increase their likelihood of becoming actively involved.

3. *Technological proficiency and resources*: Inclusive citizen science projects should consider participants' varying levels of technological proficiency and provide necessary resources and support. The findings highlight the role of technological proficiency and access to resources in facilitating individuals' participation as citizen scientists. Projects should consider participants' technological capabilities, provide necessary resources, and ensure accessibility to technology platforms. This will enable individuals to actively engage with project activities and contribute effectively to citizen science initiatives.

4. *Bridging the intention-engagement gap*: Inclusivity involves addressing the intention-engagement gap for all participants. The study reveals a significant gap between individuals' intention to participate and their actual engagement as citizen scientists. Future projects should focus on understanding the underlying factors that influence this gap and develop strategies to bridge it. By addressing barriers and creating supportive environments, citizen science initiatives can enhance participants' commitment and transform their intention into active involvement.

5. *Culture-specific engagement strategies*: Inclusivity encompasses cultural and ethnic diversity. To ensure inclusivity, citizen science projects should develop multi-lingual and language-specific engagement strategies and resources. This may involve translating materials, providing multilingual support, and actively involving participants from different cultural backgrounds in project design and decision-making processes. By embracing and valuing cultural diversity, projects can create an inclusive environment where all participants can contribute effectively.

In conclusion, this study introduced an empirical model for inclusive engagement that contributes to our understanding of the factors influencing individuals' intention and actual participation as citizen scientists while promoting the ten principles of citizen science inclusion formulated by ECSA. The findings provide valuable insights for designing inclusive recruitment strategies, promoting positive learning experiences, addressing technological barriers, bridging the intention-engagement gap, and tailoring engagement strategies to ensure cultural diversity. By implementing these implications and adhering to the principles of inclusivity, future citizen science projects can foster greater engagement and participation, ultimately leading to more impactful and inclusive scientific endeavors.

Although the research provides valuable insights into factors influencing citizens' intention and participation as citizen scientists, it is essential to acknowledge its limitations. The study's reliance on self-selected survey completions after the training phase may introduce self-selection bias, potentially influencing the results. Additionally, the focus on major native language groups might overlook other ethnic communities, affecting inclusivity. Cross-cultural differences and the timeframe of data collection could also impact the findings. Furthermore, inherent measurement limitations and contextual specificity of the CSI-COP project should be considered when interpreting the results. The sample size of 540 participants, while suitable for the project, limits the generalizability of the findings, and uneven distribution across languages raises concerns about representation. Relying on self-reported data may introduce

biases and inaccuracies due to social desirability bias. Future research should address these limitations to enhance our understanding of inclusive citizen science participation and improve the design and implementation of such initiatives.

## Supporting information

**S1 File.**
(DOCX)

**S2 File.**
(DOCX)

**S3 File.**
(PDF)

## Acknowledgments

The authors thank the citizen scientists contributing to this research.

## Author Contributions

**Conceptualization:** Shlomit Hadad, Maayan Zhitomirsky-Geffet, Huma Shah, Ulrico Celentano, Henna Tiensuu, Juha Röning, Jordi Vallverdú, Eva Jove Csabella, Olga Stepankova, John Gialelis, Konstantina Lantavou, Tiberius Ignat, Giacomo Masone, Jaimz Winter.

**Data curation:** Shlomit Hadad, Maayan Zhitomirsky-Geffet, Huma Shah, Dorottya Rigler, Ulrico Celentano, Henna Tiensuu, Juha Röning, Jordi Vallverdú, Eva Jove Csabella, Olga Stepankova, John Gialelis, Tiberius Ignat, Jaimz Winter.

**Formal analysis:** Shlomit Hadad.

**Funding acquisition:** Huma Shah.

**Investigation:** Shlomit Hadad, Maayan Zhitomirsky-Geffet, Huma Shah, Dorottya Rigler, Ulrico Celentano, Henna Tiensuu, Juha Röning, Jordi Vallverdú, Eva Jove Csabella, Olga Stepankova, John Gialelis, Konstantina Lantavou, Tiberius Ignat, Giacomo Masone, Jaimz Winter, Marica Dumitrasco.

**Methodology:** Shlomit Hadad, Huma Shah, Dorottya Rigler.

**Project administration:** Huma Shah.

**Resources:** Huma Shah.

**Supervision:** Huma Shah.

**Writing – original draft:** Shlomit Hadad, Maayan Zhitomirsky-Geffet.

**Writing – review & editing:** Maayan Zhitomirsky-Geffet, Huma Shah, Dorottya Rigler, Ulrico Celentano, Henna Tiensuu, Juha Röning, Jordi Vallverdú, Eva Jove Csabella, Olga Stepankova, John Gialelis, Konstantina Lantavou, Tiberius Ignat.

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
