## [Decision Letter · Decision Letter 0]

2 Oct 2023

PONE-D-23-25432Modeling intrinsicfactors of inclusive engagement in Citizen Science: Insights from the participants’ survey analysis of CSI-COPPLOS ONE

Dear Dr. Hadad,

Thank you for submitting your manuscript to PLOS ONE. After careful consideration, we feel that it has merit but does not fully meet PLOS ONE’s publication criteria as it currently stands. Therefore, we invite you to submit a revised version of the manuscript that addresses the points raised during the review process.

The research study is a good attempt to explore the relationship between prospective citizen scientists' traits and their engagement in citizen science (CS) projects. However, there are several key areas in the abstract and the study itself that require attention:

The abstract lacks specific details about the statistical techniques and tools used in data analysis. It is essential to provide transparency about the analytical methods employed. Moreover, the discussion of practical implications should be more extensive and explicit.

There are wide variations in demographic variables among different groups. The authors should justify how these variations impact the overall analysis. Clarification and justification of this aspect are necessary.

Table 2, The use of Pearson's correlation with nominal variables is incorrect. Pearson's correlation is suitable for continuous variables, not nominal or ordinal variables. The methodology for assessing associations among these variables should be clarified.

Table 4 the use of one-way ANOVA and post hoc tests is valid, but it's essential to explain how the effectiveness of different languages on the study variables was confirmed statistically. This can only explain and confirm the significant difference in different languages. Authors should clarify on the statistical tests and results and how it calculates the effectiveness?

Results of logistic regression needs a revisit. Table 2 shows high collinearity in the variables, The presence of multicollinearity can lead to unreliable coefficient estimates and affect the interpretation of results. It can be seen that your most of IVs are insignificant and high multicollinearity can be one of the reasons. Additionally providing information on the percent of cases correctly predicted by the model is essential for assessing its predictive power.

We look forward to receiving your revised manuscript.

Kind regards,

Prabhat Mittal, Ph.D.

Academic Editor

PLOS ONE

“The research presented in this article was part of the CSI-COP project that has received funding from the European Union’s Horizon 2020 research and innovation programme under grant agreement Nº873169.”

The authors thank the citizen scientists contributing to this research.

“This communication is part of a project that has received funding from the European Union's Horizon 2020 research and innovation program (under grant agreement Nº873169).

Initials of the authors who received each award; H.S.

5. Please amend either the title on the online submission form (via Edit Submission) or the title in the manuscript so that they are identical.

6. Please remove your figures from within your manuscript file, leaving only the individual TIFF/EPS image files, uploaded separately. These will be automatically included in the reviewers’ PDF.

Additional Editor Comments:

The research study is a good attempt to explore the relationship between prospective citizen scientists' traits and their engagement in citizen science (CS) projects. However, there are several key areas in the abstract and the study itself that require attention:

The abstract lacks specific details about the statistical techniques and tools used in data analysis. It is essential to provide transparency about the analytical methods employed. Moreover, the discussion of practical implications should be more extensive and explicit.

There are wide variations in demographic variables among different groups. The authors should justify how these variations impact the overall analysis. Clarification and justification of this aspect are necessary.

Table 2, The use of Pearson's correlation with nominal variables is incorrect. Pearson's correlation is suitable for continuous variables, not nominal or ordinal variables. The methodology for assessing associations among these variables should be clarified.

Table 4 the use of one-way ANOVA and post hoc tests is valid, but it's essential to explain how the effectiveness of different languages on the study variables was confirmed statistically. This can only explain and confirm the significant difference in different languages. Authors should clarify on the statistical tests and results and how it calculates the effectiveness?

Results of logistic regression needs a revisit. Table 2 shows high collinearity in the variables, The presence of multicollinearity can lead to unreliable coefficient estimates and affect the interpretation of results. It can be seen that your most of IVs are insignificant and high multicollinearity can be one of the reasons. Additionally providing information on the percent of cases correctly predicted by the model is essential for assessing its predictive power.

Reviewers' comments:

Reviewer's Responses to Questions

**Comments to the Author**

1. Is the manuscript technically sound, and do the data support the conclusions?

Reviewer #1: Yes

Reviewer #2: Yes

2. Has the statistical analysis been performed appropriately and rigorously? 

Reviewer #1: Yes

Reviewer #2: Yes

3. Have the authors made all data underlying the findings in their manuscript fully available?

Reviewer #1: Yes

Reviewer #2: Yes

4. Is the manuscript presented in an intelligible fashion and written in standard English?

Reviewer #1: Yes

Reviewer #2: Yes

5. Review Comments to the Author

Reviewer #1: • the paper subject " Modeling intrinsicfactors of inclusive engagement in Citizen Science: Insights from the participants’ survey analysis of CSI-COP" is significant.

• The authors claim properly placed in the context of the previous literature. This paper presents a new inclusive citizen science engagement model based on quantitative analysis of surveys administered to 540 participants of the dedicated free informal education course ‘Your Right to Privacy Online’ (MOOC - a massive online open course) from eight countries in the EU funded project, CSI-COP (Citizen Scientists Investigating Cookies and App GDPR compliance).

• But the authors need to retreat the introduction and support this research in this context. What is the difference between "Introduction" and "The objectives of CSI-Cop", the authors may have merged this to "Introduction"?

• "Theoretical framework of the study" the ten principles of CS needs revision and clear enough as a framework for this study.

• the manuscript well organized and written clearly enough to be accessible to non-specialists.

Reviewer #2: The authors in this paper introduced a new inclusive citizen science engagement model based on quantitative analysis of surveys administered to 540 participants of the dedicated free informal education course ‘Your Right to Privacy Online’ (MOOC - a massive online open course) from eight countries in the EU funded project, CSI-COP (Citizen Scientists Investigating Cookies and App GDPR compliance). The devised model offers valuable insights for designing inclusive recruitment strategies, fostering positive learning experiences, addressing technological barriers, bridging the intention-engagement gap, and tailoring engagement strategies to accommodate ethnic and cultural diversity.

I personally found this study and the obtained results quite interesting. I recommend this paper for publication in the PLOS ONE journal. The authors are suggested to add the following papers to complete the reference list:

1. Siriwardhanaa W.S.N, Rathnayakab R.M.S.S. (2022). Awareness of Counseling Psychology and the Significance of Counseling Service for the Graduate Studies, Dera Natung Government College Research Journal, 7, 70-75. DOI: https://doi.org/10.56405/dngcrj.2022.07.01.07

2. Haokip A.D., Saroh, T. 2019. Counselling Needs of Secondary School Students and Their Learning Disorders, Dera Natung Government College Research Journal, 4, 1-6. DOI: https://doi.org/10.56405/dngcrj.2019.04.01.01

3. Tiwari, S. (2023). Sustainable HRM Goals in Selected IT Companies. VEETHIKA-An International Interdisciplinary Research Journal, 9(2), 14-18. https://doi.org/10.48001/veethika.2023.09.02.003

4. Verma, R., & Bharti, U. (2023). Organizational Stress in India’s Educational Sector. VEETHIKA-An International Interdisciplinary Research Journal, 9(2), 26-29. https://doi.org/10.48001/veethika.2023.09.02.005

6. PLOS authors have the option to publish the peer review history of their article (what does this mean?). If published, this will include your full peer review and any attached files.

Reviewer #1: No

Reviewer #2: No

---

## [Author Response · Author response to Decision Letter 0]

26 Oct 2023

October, 18, 2023

To:

Dr. Prabhat Mittal, Academic Editor 

PLOS ONE

Dear Dr. Prabhat Mittal, 

Thank you for the opportunity to improve our manuscript “Modeling intrinsic factors of inclusive engagement in Citizen Science: Insights from the participants’ survey analysis of CSI-COP” (PONE-D-23-25432). We have revised the manuscript according to the editor’s and the reviewers' comments. Please find our detailed responses in the next pages. Changes are highlighted in the revised manuscript file. 

We appreciate the reviewers’ thorough review and constructive comments, as they have significantly contributed to the overall quality and rigor of our research. If any further questions or concerns arise during the review process, we remain committed to providing thorough and satisfactory clarifications. Once again, we express our gratitude for the valuable input, and we believe that the updated version of the paper addresses the reviewer's suggestions in a comprehensive and informative manner. We hope that the attached revised version of the manuscript will be considered for publication at PLOS ONE.

Sincerely,

The authors. 

Comment

The research study is a good attempt to explore the relationship between prospective citizen scientists' traits and their engagement in citizen science (CS) projects. However, there are several key areas in the abstract and the study itself that require attention:

The abstract lacks specific details about the statistical techniques and tools used in data analysis. It is essential to provide transparency about the analytical methods employed. Moreover, the discussion of practical implications should be more extensive and explicit.

Response

Following the editor's valuable comment, we have addressed the concerns regarding the abstract. Specifically, we included a reference to the statistical techniques used to enhance transparency about our analytical methods. Additionally, we expanded and made the practical implications more explicit, ensuring a comprehensive discussion of the practical applications of our findings.

Comment

There are wide variations in demographic variables among different groups. The authors should justify how these variations impact the overall analysis. Clarification and justification of this aspect are necessary.

Response

The diversity in demographic variables among different groups lies at the heart of our research endeavor. We intentionally emphasized these variations starting from the abstract, where we outlined our primary objective: exploring prospective citizen scientists' traits as intrinsic factors to enhance diversity and engagement in citizen science (CS) initiatives. This focus guided our extensive literature review, where we meticulously examined existing studies to contextualize and validate the significance of demographic variations within the realm of citizen science.

Moreover, our findings, discussion, conclusions and implications chapters provide a detailed analysis of these variations. We critically justify how diverse demographic backgrounds impact our analysis, ensuring a nuanced understanding of the data. We delve into the complexities, considering variables such as age, gender, culture, education, internet accessibility, and app usage, meticulously evaluating their influence on citizens' active engagement in CS projects. This thorough examination is crucial not only for academic rigor but also for the practical application of our research findings.

In essence, our approach is comprehensive and methodical, enabling us to thoroughly justify the impact of demographic variations on our overall analysis. We believe that our study contributes significantly to the understanding of these variations and their implications for citizen science initiatives.

 Comment

Table 2, The use of Pearson's correlation with nominal variables is incorrect. Pearson's correlation is suitable for continuous variables, not nominal or ordinal variables. The methodology for assessing associations among these variables should be clarified.

Response

We sincerely appreciate the editor's insightful feedback. In response to this comment, we have taken significant measures to rectify the issue. Specifically, we have clarified and explicitly outlined the types of tests utilized in the Statistical Data Analysis section within the Method chapter. Furthermore, we have meticulously edited and corrected Table 2, ensuring accurate representation of the correlation data.

To address the concern about using Pearson's correlation with nominal variables, we have adjusted the type of variable for the test. In our dataset, there are dichotomous nominal variables (coded as 0 and 1). For testing relationships involving these variables, we employed the Point-Biserial Correlation, which is specifically designed for such cases and is a specialized version of the Pearson correlation coefficient. Additionally, recognizing that some variables are on an ordinal scale, with levels arranged logically as detailed in Table 1, we incorporated Kendall's coefficient of rank correlation (tau-sub-b) test. This approach ensures a more accurate assessment of associations among these variables (see: Cureton, 1966; Demirtas & Hedeker, 2016; Khamis, 2008).

We are thankful for the editor's astute observation, which has significantly enhanced the accuracy of our methodology.

Cureton, E. E. (1966). Corrected item-test correlations. Psychometrika, 31(1), 93-96.

Demirtas, H., & Hedeker, D. (2016). Computing the point-biserial correlation under any underlying continuous distribution. Communications in Statistics-Simulation and Computation, 45(8), 2744-2751.

Khamis, H. (2008). Measures of association: How to choose?. Journal of Diagnostic Medical Sonography, 24(3), 155-162.

Comment

Table 4 the use of one-way ANOVA and post hoc tests is valid, but it's essential to explain how the effectiveness of different languages on the study variables was confirmed statistically. This can only explain and confirm the significant difference in different languages. Authors should clarify on the statistical tests and results and how it calculates the effectiveness?

Response

We thank the editor for the above important comment. In response, we have provided a detailed explanation of the findings and the statistical tests in both Tables 3 and 4, which is highlighted in a specific paragraph on page 22. We opted for two separate tables due to the distinct types of variables involved. Specifically, Table 3 features chi-square tests, whereas Table 4 includes ANOVA tests.

Following each table, we diligently presented and interpreted the findings. Following Table 4, significant results were further explored through pairwise comparisons. It's important to note that our findings chapter presents the results comprehensively, while the discussion chapter, spanning pages 32-33 (as indicated), delves deeply into these findings. Within this section, we extensively discussed the nuanced influence of participants' cultures, ethnicities, learning preferences, and various socio-demographic aspects on their intentions and actual engagement as citizen scientists. The discussion underscores the critical importance of tailoring engagement strategies to diverse cultural and ethnic groups, emphasizing the need for culture-specific approaches to ensure inclusivity and foster active participation in future citizen science projects. We trust these clarifications enhance the clarity and depth of our manuscript. 

Comment

Results of logistic regression needs a revisit. Table 2 shows high collinearity in the variables, the presence of multicollinearity can lead to unreliable coefficient estimates and affect the interpretation of results. It can be seen that your most of IVs are insignificant and high multicollinearity can be one of the reasons. Additionally, providing information on the percent of cases correctly predicted by the model is essential for assessing its predictive power.

Response

We sincerely appreciate the editor's detailed evaluation of our logistic regression results. We would like to clarify that collinearity indicates a linear association between two predictors, whereas multicollinearity occurs when two or more predictors are highly linearly related. In our case, we examined a correlation matrix (Table 2) where the highest significant correlation was 0.356, indicating no severe multicollinearity issues among our variables.

In response to your comment, we conducted additional tests using the variance inflation factor (VIF) metric, a reliable method for detecting multicollinearity. The VIF values ranged from 1.083 to 2.037, well below the threshold of 5, indicating moderate correlations, the absence of collinearity, and ensuring the reliability of our regression model. According to this important comment we added the following paragraph to our revised manuscript: "To assess potential multicollinearity among the independent variables, the Variance Inflation Factor (VIF) technique was employed. VIF values, ranging from 1.083 to 2.037, comfortably remained below the critical threshold of 5. These values indicate moderate correlations, affirming the robustness and reliability of the regression model." (page 26)

Furthermore, we appreciate your attention to the predictive power of our model. We provided this information in the manuscript before the findings table (Table 6 on page 26), where it was stated that " The logistic regression model was statistically significant, χ2(14) = 29.639, p = .009. The model explained 33.0% (Nagelkerke R2) of the variance in becoming a citizen scientist, and correctly classified 71.5% of cases." (page 26). We hope this additional context clarifies the robustness and predictive efficacy of our logistic regression analysis.

Comment

Response

In adherence to the specified requirement, the revised manuscript has been meticulously modified to align with PLOS ONE's style requirements.

Comment

Response

We sincerely appreciate the editor's attention to the crucial matter of participant consent in our study. We have duly addressed this concern in our methodology chapter, ensuring detailed coverage of the ethical procedures involved. Specifically, we have added a reference at the beginning of the methodology chapter (page 12, highlighted in yellow), directing readers to the consent forms and ethics applications.

Our approach to participant consent was meticulous and well-structured. Each participant, who was confirmed to be over 18 years old, provided their consent by filling out the participant informed consent form by hand, signifying their voluntary participation in the study. They were explicitly informed of the voluntary nature of their participation and the option to withdraw at any point. The informed consent process was comprehensive, including clear instructions provided to participants. They were informed about their right to withdraw their data from the project up to the data destruction date, June 30, 2032, without providing any reason, and this decision would not have any adverse effects on them.

Moreover, participants were provided with detailed information about the study procedures, including the questions they would be asked, the modes of data collection, and the expected duration. The confidentiality and data protection measures, aligning with GDPR and the Data Protection Act 2018, were meticulously explained. Personal data were anonymized, and stringent security protocols were in place for both electronic and paper records.

To ensure transparency and compliance, we maintained a clear separation between consent information and the research data, enhancing security and minimizing risks associated with data breaches. The responsibility for data destruction, in accordance with our outlined protocols, rests with the lead researcher, with all data set to be destroyed on or before June 30, 2032, ten years after the conclusion of CSI-COP.

We hope these additional details provide the necessary clarity on our ethical procedures and participant consent. Thank you for your guidance, which has significantly enhanced the integrity of our study.

Comment

“The research presented in this article was part of the CSI-COP project that has received funding from the European Union’s Horizon 2020 research and innovation programme under grant agreement Nº873169.”

The authors thank the citizen scientists contributing to this research.

“This communication is part of a project that has received funding from the European Union's Horizon 2020 research and innovation program (under grant agreement Nº873169).

Initials of the authors who received each award; H.S.

Response

In compliance with the stipulated requirement, we have meticulously removed all content pertaining to funding from the manuscript. The modified statements provided above have been incorporated into our cover letter.

Comment

Response

Following the data availability requirement, the dataset and supporting information are now made public accessible from:

 Huma Shah — Coventry University

Zenodo: https://zenodo.org/records/10043687

It will also be uploaded to CSI-COP website page for D4.3 here:

https://csi-cop.eu/project-results/who-are-citizen-scientists/

and here: https://csi-cop.eu/projectresults/

Comment

5. Please amend either the title on the online submission form (via Edit Submission) or the title in the manuscript so that they are identical.

Response

Thank you for your attention. We corrected the title in the online submission form.

Comment

6. Please remove your figures from within your manuscript file, leaving only the individual TIFF/EPS image files, uploaded separately. These will be automatically included in the reviewers’ PDF.

Response

We have removed your figures from the manuscript file. We have only left the individual image files uploaded separately.

Review Comments to the Author

Reviewer #1: 

Comment

• the paper subject " Modeling intrinsicfactors of inclusive engagement in Citizen Science: Insights from the participants’ survey analysis of CSI-COP" is significant.

• The authors claim properly placed in the context of the previous literature. This paper presents a new inclusive citizen science engagement model based on quantitative analysis of surveys administered to 540 participants of the dedicated free informal education course ‘Your Right to Privacy Online’ (MOOC - a massive online open course) from eight countries in the EU funded project, CSI-COP (Citizen Scientists Investigating Cookies and App GDPR compliance).

• But the authors need to retreat the introduction and support this research in this context. What is the difference between "Introduction" and "The objectives of CSI-Cop", the authors may have merged this to "Introduction"?

Response

We thank the reviewer for the insightful feedback. In response to the comment, we have carefully restructured the introductory chapter of our manuscript (pages 4-5). The objectives of CSI-Cop are now introduced following a concise overview of the research field, providing context for our study. This adjustment has created a more seamless transition within the introduction, integrating the objectives of CSI-Cop as an essential component. Following this introduction, the research questions are delineated, ensuring a logical and coherent progression. The reviewer's input significantly contributed to enhancing the overall flow and clarity of our manuscript.

Comment

• "Theoretical framework of the study" the ten principles of CS needs revision and clear enough as a framework for this study.

Response

We appreciate the reviewer's feedback and have carefully reviewed the theoretical framework section in light of the attached paragraph. To provide a clearer explanation of how the ten principles of CS serve as the framework for our study, we have revised the relevant section (pages 14-15). We have elaborated on how each principle informs our research design, methodology, and interpretation of results. This revised explanation aims to elucidate the direct connection between the principles of CS and the core foundations of our study, ensuring a more robust and comprehensible theoretical framework for our readers. We believe this enhancement addresses the reviewer's concerns and strengthens the theoretical foundation of our work.

Comment

• the manuscript well organized and written clearly enough to be accessible to non-specialists.

Response

We sincerely appreciate your positive feedback and recommendation for the publication of our study in PLOS ONE.

Reviewer #2: 

The authors in this paper introduced a new inclusive citizen science engagement model based on quantitative analysis of surveys administered to 540 participants of the dedicated free informal education course ‘Your Right to Privacy Online’ (MOOC - a massive online open course) from eight countries in the EU funded project, CSI-COP (Citizen Scientists Investigating Cookies and App GDPR compliance). The devised model offers valuable insights for designing inclusive recruitment strategies, fostering positive learning experiences, addressing technological barriers, bridging the intention-engagement gap, and tailoring engagement strategies to accommodate ethnic and cultural diversity. I personally found this study and the obtained results quite interesting. I recommend this paper for publication in the PLOS ONE journal. 

Response

We sincerely appreciate your positive feedback and recommendation for the publication of our study in PLOS ONE. We are delighted that you found our research and results intriguing.

Comment

The authors are suggested to add the following papers to complete the reference list:

1. Siriwardhanaa W.S.N, Rathnayakab R.M.S.S. (2022). Awareness of Counseling Psychology and the Significance of Counseling Service for the Graduate Studies, Dera Natung Government College Research Journal, 7, 70-75. DOI: https://doi.org/10.56405/dngcrj.2022.07.01.07

2. Haokip A.D., Saroh, T. 2019. Counselling Needs of Secondary School Students and Their Learning Disorders, Dera Natung Government College Research Journal, 4, 1-6. DOI: https://doi.org/10.56405/dngcrj.2019.04.01.01

3. Tiwari, S. (2023). Sustainable HRM Goals in Selected IT Companies. VEETHIKA-An International Interdisciplinary Research Journal, 9(2), 14-18. https://doi.org/10.48001/veethika.2023.09.02.003

4. Verma, R., & Bharti, U. (2023). Organizational Stress in India’s Educational Sector. VEETHIKA-An International Interdisciplinary Research Journal, 9(2), 26-29. https://doi.org/10.48001/veethika.2023.09.02.005

Response

We sincerely appreciate the reviewer's thoughtful suggestions. We diligently reviewed each of the recommended articles, but unfortunately, they did not align with the focus and scope of our research. Our study centers on adult participants above the age of 18 engaged in citizen science activities, specifically investigating cookies and App GDPR compliance.

The first suggested article focuses on school students and their experiences within a specific educational curriculum, which diverges significantly from our study involving adult citizen scientists. The second article delves into counseling needs of secondary school students, again outside the scope of our adult-focused citizen science research. The third article centers on human resource management, a topic unrelated to our investigation of online privacy and citizen science engagement. Lastly, the fourth article explores factors impacting employees in India's educational sector, a context distinct from our research involving citizen scientists.

We genuinely appreciate the reviewer's intention to enrich our literature review and assure you that our revised manuscript comprehensively covers the relevant literature within the scope of our study.

---

## [Editor Report · Decision Letter 1]

5 Nov 2023

Modeling intrinsic factors of inclusive engagement in Citizen Science: Insights from the participants’ survey analysis of CSI-COP

PONE-D-23-25432R1

Dear Dr. Hadad,

We’re pleased to inform you that your manuscript has been judged scientifically suitable for publication and will be formally accepted for publication once it meets all outstanding technical requirements.

Kind regards,

Prabhat Mittal, Ph.D.

Academic Editor

PLOS ONE
---

## [Editor Report · Acceptance letter]

15 Nov 2023

PONE-D-23-25432R1 

Modeling intrinsic factors of inclusive engagement in Citizen Science: Insights from the participants’ survey analysis of CSI-COP 

Dear Dr. Hadad:

I'm pleased to inform you that your manuscript has been deemed suitable for publication in PLOS ONE. Congratulations! Your manuscript is now with our production department. 

Kind regards, 

on behalf of

Dr. Prabhat Mittal 

Academic Editor

PLOS ONE